# Neural Attentive Circuits

**Martin Weiss**[*,1,6]   **Nasim Rahaman**[*,1,2]   **Francesco Locatello**[3]   **Chris Pal** [1,6,7]
**Yoshua Bengio**[1,5,7]   **Bernhard Schölkopf**[2]   **Nicolas Ballas** [†,4]   **Li Erran Li**[†,3]

[*,†] Equal contribution, random order.

[1]Mila, Quebec AI Institute   [2] Max Planck Institute for Intelligent Systems, Tübingen
[3] AWS AI   [4] Meta AI   [5] Université de Montréal   [6] Polytechnique Montréal
[7] Canada CIFAR AI Chair

## Abstract

Recent work has seen the development of general purpose neural architectures that can be trained to perform tasks across diverse data modalities. General purpose models typically make few assumptions about the underlying data-structure and are known to perform well in the large-data regime. At the same time, there has been growing interest in modular neural architectures that represent the data using sparsely interacting modules. These models can be more robust out-of-distribution, computationally efficient, and capable of sample-efficient adaptation to new data. However, they tend to make domain-specific assumptions about the data, and present challenges in how module behavior (i.e., parameterization) and connectivity (i.e., their layout) can be jointly learned. In this work, we introduce a general purpose, yet modular neural architecture called Neural Attentive Circuits (NACs) that jointly learns the parameterization and a sparse connectivity of neural modules without using domain knowledge. NACs are best understood as the combination of two systems that are jointly trained end-to-end: one that determines the module configuration and the other that executes it on an input. We demonstrate qualitatively that NACs learn diverse and meaningful module configurations on the Natural Language and Visual Reasoning for Real (NLVR2) dataset without additional supervision. Quantitatively, we show that by incorporating modularity in this way, NACs improve upon a strong non-modular baseline in terms of low-shot adaptation on CIFAR and Caltech-UCSD Birds dataset (CUB) by about 10 percent, and OOD robustness on Tiny ImageNet-R by about 2.5 percent. Further, we find that NACs can achieve an 8x speedup at inference time while losing less than 3 percent performance. Finally, we find NACs to yield competitive results on diverse data modalities spanning point-cloud classification, symbolic processing and text-classification from ASCII bytes, thereby confirming its general purpose nature.

## 1   Introduction

General purpose neural models like Perceivers [29] do not make significant assumptions about the underlying data-structure of the input and tend to perform well in the large-data regime. This enables the application of the same model on a variety of data modalities, including images, text, audio, point-clouds, and arbitrary combinations thereof [29, 28]. This is appealing from an ease-of-use perspective, since the amount of domain-specific components is minimized, and the resulting models can function well *out-of-the-box* in larger machine learning pipelines, e.g., AlphaStar [28].

At the same time, natural data generating processes can often be well-represented by a system of sparsely interacting independent mechanisms [41, 46], and the Sparse Mechanism Shift hypothesis

36th Conference on Neural Information Processing Systems (NeurIPS 2022).

stipulates that real-world shifts are often sparse when decomposed, i.e., that most mechanisms may remain invariant [47]. Modular architectures seek to leverage this structure by enabling the learning of systems of sparsely interacting neural modules [3, 20, 44, 45]. Such systems tend to excel in low-data regimes, systematic generalization, fast (sample-efficient) adaptation to new data and can maintain a larger degree of robustness out-of-distribution [52, 3, 4, 21]. However, many of these models make domain-specific assumptions about the data distribution and modalities [4] – e.g., Mao et al. [37] relying on visual scene graphs, or Andreas et al. [3] using hand-specified modules and behavioral cloning. As a result, such models are often less flexible and more difficult to deploy.

In this work, we propose Neural Attentive Circuits (NACs), an architecture that incorporates modular inductive biases while maintaining the versatility of general purpose models. NACs implement a system of sparsely and attentively interacting neural modules that pass messages to each other along *connectivity graphs* that are learned or dynamically inferred in the forward pass and subject to priors derived from network science [25, 6, 18]. NACs demonstrate strong low-shot adaptation performance and better out-of-distribution robustness when compared to other general purpose models such as Perceiver IOs [28]. Besides scaling linearly with input size (like Perceivers), we show that by pruning modules, the computational complexity of NACs can be reduced at inference time while preserving performance. In addition, we propose a conditional variant where the graph structure and module parameterization is conditioned on the input. This enables conditional computation [10, 9], where the functions computed by neural modules and the sparse connectivity between modules is determined only at inference time. A qualitative analysis on Natural Language and Visual Reasoning for Real (NLVR2) shows that conditional NACs can learn connectivity graphs that are meaningfully diverse.

Figure 1 shows the schematics of the proposed architecture, with its two core components: the **circuit generator** and the **circuit executor**. The circuit generator produces the configuration over modules, which we call a circuit design, defining (a) the connectivity pattern between the neural modules, and (b) instructions that condition the computation performed by each module. The circuit design may either be conditioned on (part of) the sample (in the case of conditional NACs), or it may simply be learned over the course of training via gradient descent (in the case of unconditional NACs). The circuit executor consumes the circuit design and an input sample to perform inference.

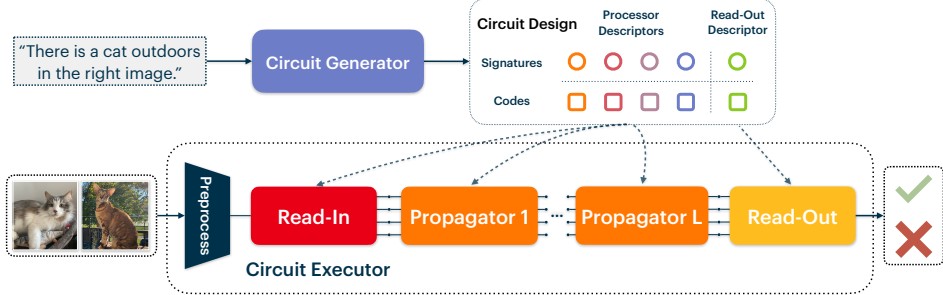

Figure 1: **Circuit Generator and Executor.** We show NAC applied to a natural language and visual reasoning task where text conditions the circuit generator and the program executor consumes images.

**Our contributions are as follows.** **(a)** We propose Neural Attentive Circuits (NACs), a general purpose neural architecture that jointly learns the module parameterization and connectivity end-to-end and gracefully supports more than one thousand sparsely interacting modules. **(b)** We demonstrate the general-purpose nature of NACs by training it on a selection of diverse data domains covering natural images, 3D point-clouds, symbolic processing, text understanding from raw ASCII bytes, and natural-language visual reasoning. **(c)** We quantitatively evaluate the out-of-distribution robustness and few-shot adaptation performance of NACs. We find that in the low-shot regime, NACs can outperform Perceiver IOs by approximately 10% on 8-Way CIFAR and CUB-2011. Further, the proposed model yields roughly 2.5% improvement on TinyImageNet-R, an out-of-distribution test-set for TinyImageNet based on ImageNet-R [24]. **(d)** We explore the adaptive-computation aspect of NACs, wherein the computational complexity of a trained model can be reduced at inference time by roughly 8 times, at the cost of less than 3% loss in accuracy on Tiny-ImageNet. **(e)** We qualitatively demonstrate that it is possible to condition the circuit design (i.e., configuration over modules) on the input. On NLVR2, we use the text modality of the sample to condition the circuit design, which is

then applied to the image modality of the same sample to produce a result. We find that connectivity graphs generated by sentences that involve similar reasoning skills have similar structures.

## 2 Neural Attentive Circuits

This section describes Neural Attentive Circuits (NACs), a neural architecture that can consume arbitrary (tokenized) set-valued inputs including images and text. NACs are a system of neural modules that learn to interact with each other (see Figure 2). Module connectivity is learned end-to-end and is subject to regularization that encourages certain patterns (e.g., sparsity, scale-freeness and formation of cliques). In the following, we first describe two key building blocks of the proposed architecture. We then introduce the circuit executor that infers an output given some input and a circuit design. We finally introduce the circuit generator which produces said circuit design.

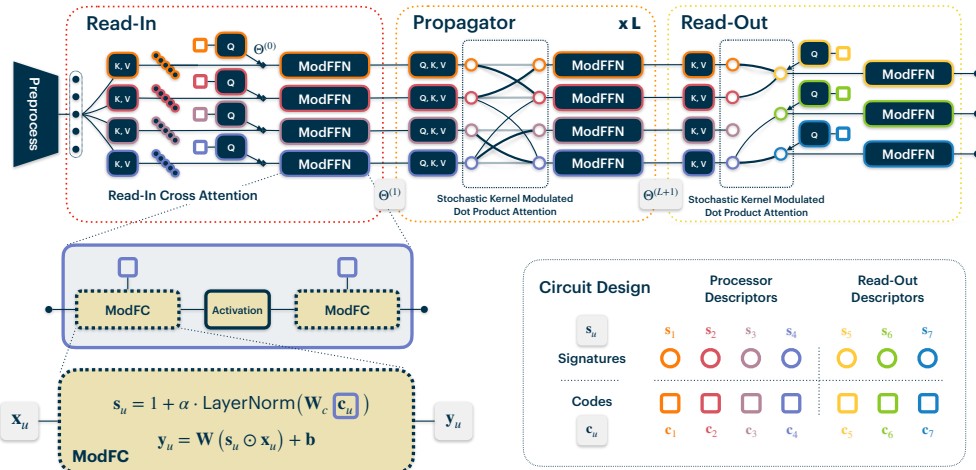

Figure 2: **Schematics of Neural Attentive Circuits.** *Top:* a NAC with $U_p = 4$ processor modules and $U_o = 3$ read-out modules. *Bottom right:* each module has a unique signature and code (colored circles and boxes). The signatures determine which modules interact and the code given to a module conditions the computation it performs (via ModFFNs and ModFCs). *Bottom left:* visual description of ModFFNs and ModFC.

**Circuit Design.** Consider a NAC (Figure 2) with $U$ modules, of which $U_p$ are *processor modules* (responsible for the bulk of the computation) and $U_o$ are *read-out modules* (responsible for extracting model outputs). By circuit design, we refer to the set of $U$ descriptors that condition the module connectivity and computation carried out at each module. Each descriptor comprises three vectors $(\boldsymbol{s}_u, \boldsymbol{c}_u, \boldsymbol{\theta}_u^{(0)})$: a signature vector $\boldsymbol{s}_u \in \mathbb{R}^{d_{sig}}$, a code vector $\boldsymbol{c}_u \in \mathbb{R}^{d_{code}}$ and a module initial state $\boldsymbol{\theta}_u^{(0)} \in \mathbb{R}^{d_{model}}$. The set of signature vectors $\{\boldsymbol{s}_u\}_u$ determines the connectivity between modules while the code vector $\boldsymbol{c}_u$ conditions the computation performed by module at index $u$. Further, the vector $\boldsymbol{\theta}_u^{(l)} \in \mathbb{R}^{d_{model}}$ tracks the state of the module $u$ before the $l$-th layer, and by $\Theta^{(l)}$ we denote the set of all such states (over modules). Each module updates its state as the computation progresses in depth (see below). Finally, processor module signatures and codes are shared across depth.

**ModFC and ModFFN.** The Modulated Fully-Connected (ModFC) layer is the component that enables a module to condition its computation on a code. It replaces the fully-connected layer and supports a multiplicative conditioning of the input vector $\boldsymbol{x}$ by the *code vector $\boldsymbol{c}$*. It can be expressed

$$\boldsymbol{y} = \mathrm{ModFC}(\boldsymbol{x}; \boldsymbol{c}) = \boldsymbol{W}\left(\boldsymbol{x} \odot (1 + \alpha \, \mathrm{LayerNorm}(\boldsymbol{W}_c \boldsymbol{c}))\right) + \boldsymbol{b}, \quad (1)$$

where $\odot$ denotes element-wise multiplication, $\boldsymbol{W} \in \mathbb{R}^{d_{out}} \times \mathbb{R}^{d_{in}}$ is a weight matrix, $\boldsymbol{b} \in \mathbb{R}^{d_{out}}$ is a bias vector, $\boldsymbol{c} \in \mathbb{R}^{d_{code}}$ is the conditioning code vector, $\boldsymbol{W}_c \in \mathbb{R}^{d_{in}} \times \mathbb{R}^{d_{code}}$ is the conditioning weight matrix, and $\alpha$ is a learnable scalar parameter that controls the amount of conditioning. Setting $\alpha = 0$ removes all conditioning on $\boldsymbol{c}$, and we generally initialize $\alpha = 0.1$. Multiple such ModFC layers (sharing the same code vector $\boldsymbol{c}$) and activation functions can be stacked to obtain a ModFFN

layer. Like in transformers, multiple copies of ModFCs and ModFFNs are executed in parallel. But unlike in transformers, each copy is conditioned by a unique learnable code vector and therefore performs a different computation. Consequently, though the computation performed in each module is different, the total number of parameters does not noticeably increase with the number of modules. See a clarifying example in Appendix B.

**SKMDPA.** Stochastic Kernel Modulated Dot-Product Attention (SKMDPA) is a sparse-attention mechanism that allows the modules to communicate. Given two modules at index $i$ and $j$, the distance between their signatures $s_i$ and $s_j$ determines the probability with which these modules can interact via dot-product attention – smaller distance implies higher likelihood that the modules are allowed to communicate. Like vanilla dot-product attention, SKMDPA operates on a set of query $q_i$, key $k_j$, and value $v_j$ vectors, where $i$ indexes queries and $j$ indexes keys and values. But, in addition, each query $q_i$ and key $k_j$ is equipped with a signature embedding vector, which may be learned as parameters or dynamically inferred. These signatures are used to sample a kernel matrix $K$ [1] via

$$K_{ij} \sim \text{Concrete}(P_{ij}, \tau) \text{ with } P_{ij} = \exp\left[\frac{-d(s_i, s_j)}{\epsilon}\right], \tag{2}$$

where $d$ is some distance metric, $\epsilon$ is a bandwidth (a hyper-parameter), Concrete is the continuous relaxation of the Bernoulli distribution [36] with sampling temperature $\tau$, and the sampling operation $\sim$ is differentiable via the reparameterization trick. $K_{ij}$ is likely to be close to 1 if the distance between $s_i$ and $s_j$ is small, and close to 0 if not. As $\tau \to 0$, the kernel matrix $K_{ij}$ becomes populated with either 1s or 0s. Further, if $P_{ij}$ is ensured to be sufficiently small on average, $K_{ij}$ is a sparse matrix. In practice, we set $\tau$ to a small but non-zero value (e.g. 0.5) to allow for exploration. Now, where $A_{i \leftarrow j}$ is the dot-product attention score $q_i \cdot k_j / \sqrt{d}$ between the query $q_i$ and key $k_j$, the net attention weight $W_{i \leftarrow j}$ and the output $y_i$ are

$$W_{i \leftarrow j} = \text{softmax}_j \left[A_{i \leftarrow j} + \log \hat{K}_{ij}\right] \text{ where } \hat{K}_{ij} = {}^{K_{ij}}/_{(\delta + \sum_j K_{ij})} \text{ and } y_i = \sum_j W_{i \leftarrow j} v_j. \tag{3}$$

Here, $W_{i \leftarrow j}$ denotes the weight of the *message* $v_j$ passed by element $j$ to element $i$, and $\delta$ is a small scalar for numerical stability. We note that if $K_{ij} \approx 0$, we have $W_{i \leftarrow j} \approx 0$, implying if $K_{ij}$ is sparse, so is $W_{i \leftarrow j}$. The proposed attention mechanism effectively allows the query $q_i$ to interact with the key $k_j$ with a probability that depends on the distance between their respective signatures. In this work, this distance is derived from the cosine similarity, i.e., $d(s_i, s_j) = 1 - \text{CosineSimilarity}(s_i, s_j)$. We note the connection to Lin et al. [34], where the attentive interactions between queries and keys are dropped at random, albeit with a probability that is not differentiably learned.

## 2.1 Circuit Executor

The circuit executor takes the circuit design and executes it on an input to make a prediction. It has four components: **(a)** a tokenizer to convert the input (e.g., images or text) to a set of vectors, **(b)** a *read-in* layer which allows the processor modules to attentively read from the inputs, **(c)** a sequence of *propagator layers* which iteratively update the processor modules states through a round of communication and computation, **(d)** *read-out* layers that enable the read-out modules to attentively poll the final outputs of the processor modules.

**Tokenizer.** The tokenizer is any component that converts the input to the overall model to a set of representation vectors (with positional encodings, where appropriate), called the input set $X$. It can be a learned neural network (e.g., the first few layers of a ConvNet), or a hard-coded patch extractor. _Role:_ The tokenizer standardizes the inputs to the model by converting the said inputs to a set of vectors.

**Read-In.** The read-in component is a read-in attention followed by a ModFFN. Read-in attention is a cross-attention that uses the initial processor module states $\Theta_p^{(0)}$ to generate a query vector and uses the inputs $X$ to generate keys and values. Unlike typical cross-attention [32, 29], we use ModFCs instead of FCs as query, key and value projectors. Where $x_j$ is an element of $X$, we have:

$$q_u = \text{ModFC}(\theta_u^{(0)}, c_u) \qquad k_{uj} = \text{ModFC}(x_j, c_u) \qquad v_{uj} = \text{ModFC}(x_j, c_u) \tag{4}$$

$$\hat{y}_u = \sum_j \frac{q_u \cdot k_{uj}}{\sqrt{d}} v_{uj} \qquad y_u = \text{ModFC}(\hat{y}_u, c_u) \qquad \theta_u^{(1)} = \text{ModFFN}(y_u, c_u) \tag{5}$$

Each query $\boldsymbol{\theta}_u^{(0)}$ sees its own set of keys $\boldsymbol{k}_{uj}$ and values $\boldsymbol{v}_{uj}$, owing to the conditioning on the code $\boldsymbol{c}_u$. The read-in attention is followed by $U_p$ copies of a two layer ModFFN each conditioned on a code $\boldsymbol{c}_u$. The read-in outputs a set of vectors $\boldsymbol{\theta}_u^{(1)} \in \Theta_p^{(1)}$. _Role:_ The read-in attends to the inputs via a cross-attention mechanism that scales linearly with the size of the input.

**Propagator Layers.** After the read-in operation, propagator layers sequentially update the state of each processor module $L$ times. Each propagator layer implements a round of module communication via SKMDPA, followed by the computation of $U_p$ copies of a ModFFN. Both operations are conditioned on their processor module descriptor codes. The $l$-th propagator (where $l = 1, \ldots, L$) ingests the set of states $\Theta_p^{(l)}$ and outputs another set $\Theta_p^{(l+1)}$ of the same size. _Role:_ A propagator layer updates each processor module state vector by applying learned stochastic attention and module computation.

**Read-Out.** The read-out component is a SKMDPA-based cross-attention mechanism (read-out attention) followed by a ModFFN, and it is responsible for extracting the output from the processor module states after the final propagator layer. Read-out modules have their own signatures and codes parameters which are different from the processor modules. The read-out attention uses the initial states $\Theta_o^{(0)}$ of these modules as queries, and the final states of the processor modules $\Theta_p^{(L+1)}$ to obtain keys and values. Finally, we note that in vanilla classification problems, we found it useful to use several read-out modules. Each read-out module produces a confidence score, which is used to weight their respective additive contribution to the final output (e.g., classification logits). _Role:_ The read-out is responsible for attending to the final processor module states to produce an overall output.

## 2.2 Circuit Generator

Recall that the circuit generator produces a circuit design that specifies the connectivity between the modules (via signatures) and the computation performed by each module (via codes). There are two ways in which the circuit generator can be instantiated, as described below.

**Unconditional Circuit Generator.** In the first variant (unconditional), the circuit design is induced by signatures $\boldsymbol{s}_u$ and codes $\boldsymbol{c}_u$ that are freely learnable parameters (where $u$ indexes the module descriptor). The initial state $\boldsymbol{\theta}_u^{(0)}$ is obtained passing $\boldsymbol{c}_u$ through a two layer MLP. We refer to the overall model as an unconditional NAC, or simply a NAC. _Role:_ The unconditional circuit generator implements a mechanism to learn the circuit design via free parameters with gradient descent.

**Regularization via Graph Priors.** In the unconditional setting, we observe a task-driven pressure that collapsed all signatures $\boldsymbol{s}_u$ to the same value (see Appendix A). This results in a graph that is fully connected, i.e., all modules interact with all other modules, thereby forgoing the benefits of sparsely interacting modules. To counteract this pressure, we impose a structural prior on the learned graph via a regularization term in the objective, which we call the Graph Structure Regularizer. The graph prior is expressed as a target link probability matrix and the extent to which the prior is satisfied is obtained comparing the closest permutation of $P_{ij}$ (the generated link probability matrix) to the target values. This regularizer encourages the graph to have certain chosen properties, while still preserving the flexibility afforded by learning the graph structure. We experiment with several such properties, including scale-freeness [6] (leading to a few well connected nodes or _hubs_, and a heavy tail of less connected nodes), and a planted partition [15] one based on the stochastic block model [25] (encouraging modules to group themselves in _cliques_, where communication within cliques is less constrained than that between cliques). In Section 4, we will find that some graph properties do indeed yield better out-of-distribution performance, whereas others perform better in-distribution. Figure 3 shows the class of graphs we experiment with in this work, and Appendix A contains the details. _Role:_ The Graph Structure Regularizer enforces sparsity by preventing the module connectivity graph from collapsing to an all-to-all connected graph, while inducing graph properties that may unlock additional performance depending on the task.

**Conditional Circuit Generator.** In the second variant (conditional), each sample is assigned a different circuit design. In this case, the circuit generator takes as input the sample $X$ (or a part thereof), and outputs a set of signatures and codes $\{\boldsymbol{s}_u(X), \boldsymbol{c}_u(X)\}_u$. This setting is particularly interesting for multi-modal data, where the circuit generator may ingest one modality and the circuit executor may consume another. In this work, we implement the conditional circuit generator as a simple cross attention mechanism, where learned parameters are used as queries (one per module) and elements of $X$ as keys and values [32]. We refer to the resulting model as conditional NACs,

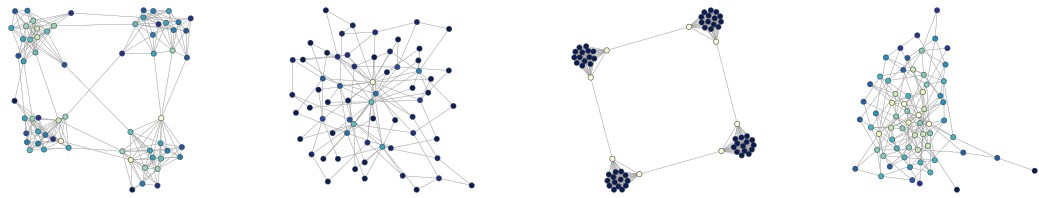

Figure 3: Samples from **Graph priors** explored here. Node colors represent degree (# of edges). **Left**: Planted-partition [15, 25]. **Center-left**: Scale-free [7, 6]. **Center-right**: Ring-of-Cliques [38]. **Right**: Erdos-Renyi [17].

while noting that the circuit generator could in principle be implemented by other means (e.g., auto-regressive models and GFLowNets [12]) in future work. *Role**:* The conditional circuit generator relies on the sample when producing a circuit design. This way, the connectivity between modules and the computation performed by each module can be dynamically changed for each sample.

**In summary,** we have introduced Neural Attentive Circuits (NACs) and its conditional variant, a general purpose neural architecture of attentively interacting modules. We discussed the two key components of NACs: **(a)** a circuit generator that *generates* a circuit design determining which modules connect to which others (via signatures) and the function performed by each (via codes), and **(b)** a circuit executor that *executes* the configuration on an input to infer an output.

## 3 Related work

**General Purpose Models.** A recent line of work develops *general-purpose models* that make few assumptions about the data domain they are applied to [29, 28, 23]. Of these models, Perceiver IO [28] is most similar to NACs, in that **(a)** they both rely on a cross-attention to both read (tokenized) set-valued inputs into the model and extract set-valued output from the model, and **(b)** their computational complexity scales linearly with the input size. However, unlike Perceiver IO (PIO), **(a)** the connectivity between NAC modules is not all-to-all, but sparse in a learned way, and **(b)** the computation performed by NAC's equivalent of a PIO latent is conditioned at all layers via the ModFC component.

**Modular Inductive Biases.** Modular neural architectures are known to yield models that are capable of systematic generalization and fast adaptation [5, 2]. Of these architectures, Neural Interpreters (NIs) are most similar to NACs, in that they both use the framework of signatures and codes, where signatures determine the computational pathway and codes condition individual modules. However, there are four key differences: **(a)** unlike in NIs, the routing of information in NACs is stochastic, **(b)** the mechanism to condition on codes (ModFC) is different, and **(c)** the computational complexity and memory requirements of NIs scales quadratically with the size of the input, whereas in NACs, this scaling is linear, and **(d)** NIs are only known to work with up to 8 modules, whereas NACs can support more than a thousand modules on a single GPU (i.e., without model parallelism).

**Neuro-symbolic Models.** Neuro-symbolic models have shown strong systematic generalization on synthetic datasets such as CLEVR [30] and SQOOP [5]. However, models of this class such as the Neuro-Symbolic Concept Learner [37] and Neural Module Networks [3] are not general-purpose, using semantic parsers and domain-specific modules. In contrast, NACs are general-purpose and can jointly learn a sparse layout and parameterization for more than one thousand homogeneous modules.

## 4 Experiments

In this section, we empirically investigate Neural Attentive Circuits (NACs) in five problem settings with different modalities: image classification, point cloud classification, symbolic processing (ListOps [49]), text classification from raw bytes [49], and multi-modal reasoning over natural language and images [48]. We compare against baselines from the literature, as well as those that we train under controlled settings. The questions we ask are the following: **(a)** Does the inductive bias of sparsely interacting modules improve NACs out-of-distribution robustness and fast (few-shot) adaptation to new data? **(b)** How important is it that these modules be capable of modelling different functions (via ModFC layers), and that the connectivity between modules is learned (via SKMDPA)? **(c)** Do the modules strongly co-adapt to each other, or can the system still function with some modules

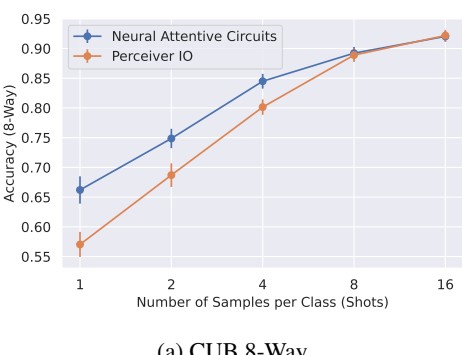
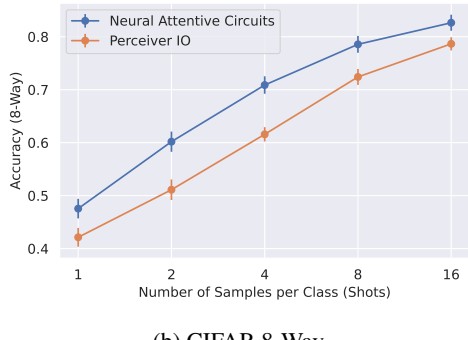

(a) CUB 8-Way               (b) CIFAR 8-Way

Figure 4: Few-Shot Adaptation Results on CUB and CIFAR. On each dataset, we fine-tune ImageNet pretrained NAC and PIO models to classify images into 8 classes and vary the number of examples ("shots") per class from 1 to 16. In the low-shot transfer regime (e.g., 1-4 samples per class), we find that NAC compares favorably with the baseline PIO, suggesting that the modular inductive biases in the former can aid fast adaptation.

pruned at inference time? **(d)** In the sample-conditional setting, are there interesting patterns in the generated circuit designs? **(e)** Are NACs general-purpose?

## 4.1 Image Classification

In this section, we conduct a quantitative evaluation of NACs on image classification problems. We use Tiny-ImageNet as our primary test-bed. Tiny-ImageNet is a miniaturized version of ImageNet[16], with smaller dataset size and lower image resolution. It allows for compute efficient experimentation, but makes the overall classification task significantly more challenging, especially for larger models that can easily overfit [33]. To evaluate out-of-distribution (OOD) robustness in this regime, we use Tiny-ImageNet-R, which is a down-sampled subset of ImageNet-R(enditions) [24]. It contains roughly 12000 samples categorized in 64 classes (a subset of Tiny-ImageNet classes), spread across multiple visual domains such as art, cartoons, sculptures, origami, graffiti, embroidery, etc. For low-shot adaptation experiments, we also train a NAC on ImageNet, in order to compare with a pretrained baseline. Additional details can be found in Appendix C.

**Baselines.** For a certain choice of hyper-parameters, NACs can simulate Perceiver IOs (PIOs) [28], making the latter the natural baseline for NACs. For all Tiny-ImageNet runs for both NACs and PIOs, we use the same convolutional preprocessor that reduces the $64 \times 64$ input image to a $8 \times 8$ feature map. For ImageNet training, we use the same convolutional preprocessors as we do for Tiny-ImageNet. Please refer to Appendix C for additional details.

**Pre-training.** We pretrain all models on Tiny-ImageNet for 400 epochs, and perform model selection based on accuracy on the corresponding validation set. We evaluate the selected models on Tiny-ImageNet-R. For few-shot adaptation experiments, we use the official ImageNet pretrained weights for a 48-layer deep PIO [28], which yields 82.1% validation accuracy. To compare few-shot adaptation performance with this model, we train a 8-layer deep NAC with a scale-free prior graph on full ImageNet for 110 epochs, which yields 77% validation accuracy with 1024 modules. Additional details and hyperparameters can be found in Appendix C.

**Few-Shot Adaptation.** Bengio et al. [11] proposed that models that learn a good modularization of the underlying data generative process should adapt in a sample-efficient manner to unseen but related data-distributions. Motivated by this reasoning, we measure NACs few-shot adaptation performance and compare with a non-modular PIO baseline. To this end, we fine-tune ImageNet pre-trained NACs and PIOs on small numbers of samples – between 8 and 128 in total – from two different datasets: CUB-2011 (Birds) and CIFAR. The results shown in Figure 4 support the hypothesis that the modular inductive biases in NACs help with sample-efficient adaptation, and we show the few-shot experiments with varying numbers of classes in the Appendix with similar results.

**Ablations.** In this experiment (Table 1), we start with a Perceiver IO and cumulatively add design elements to arrive at the NAC model. We evaluate the IID and OOD performances and make two key observations. **(a)** We find that adding ModFC layers does not significantly affect IID generalization, but improves OOD generalization. Moreover, using a static Barabasi-Albert attention graph (via frozen signatures) reduces the IID performance while further improving OOD performance. Jointly

learning the connectivity graph and module codes yields the largest improvements. Learning the graph allows to recover the IID performance, and the OOD performance is further improved. **(b)** The choice of a graph prior exposes a trade-off between IID and OOD performance. The scale-free prior yields the best IID performance, whereas the ring-of-cliques prior yields the best OOD performance.

| | IID (Tiny-ImageNet Validation) | | OOD (Tiny-ImageNet-R) | |
|---|---|---|---|---|
| | Acc@1 | Acc@5 | Acc@1 | Acc@5 |
| Perceiver IO | 58.89 | 80.09 | 17.57 | 37.16 |
| + ModFC | 58.91 | 79.43 | 18.01 | 37.05 |
| + Sparse graph (not learnable) | 58.31 | 80.06 | 18.50 | 37.92 |
| NAC with Scale-Free Prior | **60.76** | 80.86 | 19.52 | 38.52 |
| NAC with Planted-Partition Prior | 60.71 | **81.34** | 19.42 | 39.84 |
| NAC with Ring-of-Cliques Prior | 60.54 | 81.18 | **20.03** | **40.44** |
| NAC with Erdos-Renyi Prior | 60.33 | 81.08 | 19.83 | 39.32 |

Table 1: **Ablation and Graph Results.** In this experiment, we start with a Perceiver IO and cumulatively add design elements to arrive at the NAC model. At each stage, we train and observe the effect on generalization. We find that adding ModFC layers alone does not have a clear impact on either IID or OOD generalization. However, OOD is then clearly improved by adding a sparse kernel sampled from a frozen prior distribution. Below the midrule, we evaluate NACs performance with several different graph prior regularizers. We see that learning the graph via SKMDPA allows the model to improve both its IID and OOD performance.

**Effect of Dropping Modules.** In this experiment, we seek to measure the extent to which the NACs modules depend on other modules to function. To this end, we drop modules in order of their connectivity (i.e., least connected modules are dropped first) and measure the performance of the model on Tiny-ImageNet along the way. We perform this experiment for models trained with different types of graph priors, and show the result in Figure 5. We make two observations: **(a)** it is possible to remove a large fraction (e.g., more than 80%) of the modules at little cost to accuracy, obtaining a $\sim 8\times$ speedup in the process, and **(b)** different graph priors offer different amounts of robustness towards dropped modules. The former observation is particularly interesting from a computational efficiency perspective (Figure 5b), since dropping modules increases throughput and decreases memory requirements at inference-time without additional implementation overhead. The latter observation suggests a way in which the choice of a graph prior can affect the properties of the NACs model. We provide additional details in Appendix F, where Figure 15 compares the effect of sparsification on NACs (with different graph priors) and Perceiver IO.

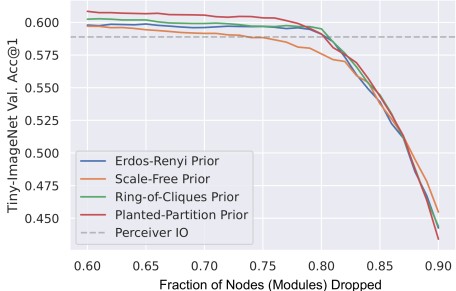 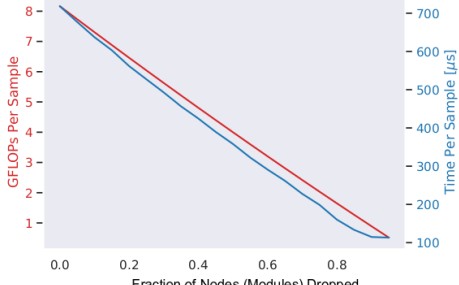

(a) Validation accuracy as modules are dropped.    (b) Inference speed as modules are dropped.

Figure 5: **Adaptive Computation**. We train models to convergence and then sparsify them at inference time and evaluate their validation accuracy. In both plots, the x-axis shows the number of modules dropped at inference time, and the Y-axis shows the accuracy of the model on Tiny-ImageNet after sparsification at inference time. We observe that the Neural Attentive Circuit (NAC) is much more robust than Perceiver IO to sparsification at inference time, and can drop over 80% of its circuits before its performance drops below Perceiver IO. As a result, NACs can be adapted at inference time to a wide range of computational budgets.

## 4.2 Point Cloud Classification

We evaluate the NAC architecture on a point cloud classification benchmark without additional data augmentation, geometric inductive biases or self-supervised pre-training [14]. Comparably, Perceiver IO obtained 77.4%, Hierarchical Perceivers achieved 75.9% without self-supervised pre-training, and when trained with self-supervised pre-training Hierarchical Perceivers obtained 80.6% (cf. Table 7 of [14]). We trained a NAC without specialized tokenization or pre-training to achieve a test accuracy of 83%, demonstrating that the sparse inductive prior was quite effective in this setting. The original Perceiver [29] relies on additional data augmentation beyond that used in [14] to obtain 85% test accuracy. We do not know how well the original Perceiver would perform without the additional augmentations, but expect it to be similar to Perceiver IO.

| Method | Test Accuracy |
| --- | --- |
| Hierarchical Perceiver (No MAE) [14] | 75.9% |
| Perceiver IO [14] | 77.4% |
| Hierarchical Perceiver (with MAE) [14] | 80.6% |
| **NAC (ours)** (No MAE) | **83.0%** |

Table 2: **Point Cloud Classification.** In this experiment, we compare against results reported in Carreira et al. [14]. In that work, the authors were able to improve over Perceiver IO results through a masked auto-encoding (MAE) pre-training step. We find that, without pre-training, NACs with a Scale-Free sparse prior graph are able to achieve superior test accuracy.

## 4.3 Symbolic Processing on ListOps and Text Classification from Raw Bytes

In this set of experiments, we evaluate the ability of NACs to reason over raw, low-level inputs. We use two tasks to this end: a symbolic processing task (ListOps) and a text-classification task from raw ASCII bytes. The former (ListOps) is a challenging 10-way classification task on sequences of up to 2,000 characters [39]. This commonly used benchmark [49] tests the ability of models to reason hierarchically while handling long contexts. The latter is a binary sentiment classification task defined on sequences of up to 4000 bytes. In particular, we stress that the models ingest raw ASCII bytes, and **not** tokenized text as common in most NLP benchmarks [49]. This tests the ability of NACs to handle long streams of low-level and unprocessed data.

The results are presented in Table 3. We find that NACs are competitive, both against a general-purpose baseline (our implementation of Perceiver IO) and other third-party baselines, including full-attention transformers and the Linformer. This confirms that NACs are indeed general purpose, and that general-purpose architectures can be effective for handling such data.

## 4.4 Sample-Conditional Circuit Design Generation

In the previous sections, we quantitatively analyzed the performance of a NAC model equipped with a connectivity graph that is learned, but fixed at inference time. In this section, we investigate sample-conditional circuit generation without regularizing the graph structure. Concretely, we study the circuit designs generated by a conditional NAC trained to perform a multi-modal reasoning task. In our experiments with conditional NACs, the circuit generator was implemented by a simple cross attention layer followed by a feed forward network (FFN), and a self-attention layer followed by

| | **Test Accuracy** | |
| --- | --- | --- |
| Method | ListOps [39] | Text Classification [49] |
| Full Attention Transformers [49] | 37.13% | 65.35% |
| Linformer [49] | 37.38% | 56.12% |
| Perceiver IO (our impl.) | 39.70% | 66.50% |
| NAC (ours) | **41.40%** | **68.18%** |

Table 3: **ListOps and Text Classification Results.** In this experiment, we train a Perceiver IO and a NAC from scratch on two tasks from the the Long Range Arena [49]: ListOps symbolic processing and text classification from ASCII bytes. We observe that NAC excels in processing long-range dependencies in this setting.

two FFNs run in parallel, see Appendix G for further details. We use Natural Language for Visual Reasoning *for Real* (NLVR2) dataset [48] which contains natural language sentences grounded in images. The task is to determine whether a natural language statement holds true for a given pair of photos. Solving this task requires reasoning about sets of objects, making comparisons, and determining spatial relations. Accordingly, we would expect that the circuit generator creates circuit designs that are qualitatively different for different logical relations.

**Training.** We condition the circuit generator on natural language captions, and the executor outputs a binary label conditioned both on the image pairs and on the output of the circuit generator. Given that our goal of understanding how conditional NACs perform reasoning, we use a (frozen) pre-trained CLIP [42] backbone to pre-process both texts and images. We select CLIP because it allows us to measure zero-shot performance out-of-the-box (53% validation accuracy, details in Appendix G), and observe that conditional NACs improve on it (obtaining approximately 64% validation accuracy).

**Analysis.** We select a trained model and analyze the behaviour of its circuit generator. In Figure 6, we visualize the 2D embeddings of the link probability matrix $P$ (see Equation 2), as well as a selection of inferred connectivity graphs for simple (non-compound) natural language statements that capture a subset of the semantic phenomena found in Suhr et al. [48] (see Appendix G). To obtain the former, we flatten the lower-triangular part of the link probability matrix $P_{ij}$ into a vector. We first reduce this ∼50k-dimensional vector to 64 dimensions via PCA [31]. Subsequently, TSNE [51] is used to embed the 64 dimensional vector in 2 dimensions. To obtain the plots of module connectivity graphs, we use a force-based graph visualization subroutine in the network analysis library NetworkX [22], wherein the attractive force between two modules is inversely proportional to the distance of their respective signatures. We draw an edge between modules if they are more than 50% likely to be connected with each other, and the module colors are consistent over all shown graphs. **Key observations: (a)** there is diversity in the generated graph in the conditional setting, even though we do not use a regularizing prior like we did in the image classification task; **(b)** different sentence types are assigned to different graph structures. Further, the generated graphs share a similar structure with two large cliques, with nodes that bridge them.

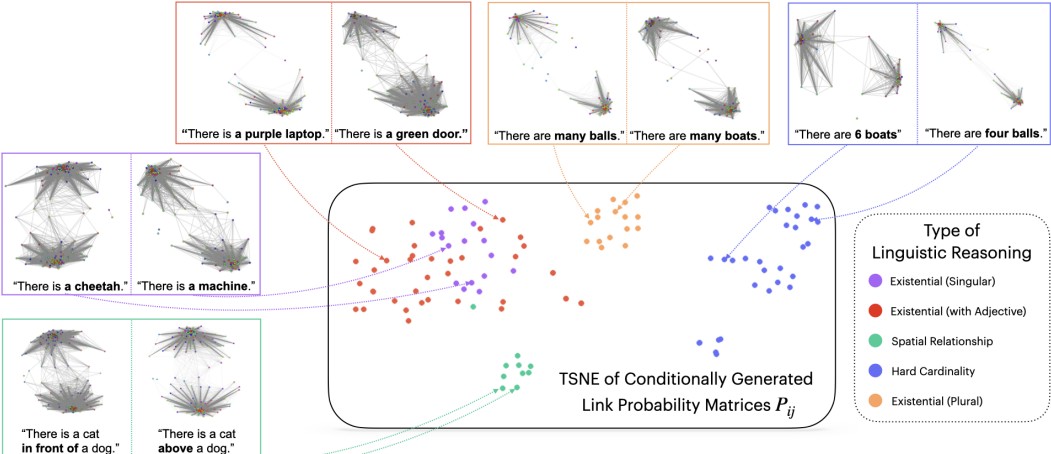

Figure 6: **Link Probability Matrix Embeddings**. This figure shows a TSNE [51] embedding of the generated graphs provided different kinds of text input. We color-code text inputs based on the type of reasoning that it requires; here we show *hard cardinality* which includes a precise number of objects, *soft cardinality* which includes words like "many" or "few", *existential* which only indicates that an object is present, and spatial relationships which use a prepositional phrase to denote a physical relationship. We see by the clustering that the low-dimensional embedding space of the generated graph captures aspects of the reasoning task. We also note that all learned graphs are similarly structured with two large cliques, but the specific arrangement varies widely.

**In conclusion,** we have shown a framework in Neural Attentive Circuits (NACs) for constructing a general-purpose modular neural architecture that can jointly learn meaningful module parameterizations and connectivity graphs. We have demonstrated empirically that this modular approach yields improvements in few-shot adaptation, computational efficiency, and OOD robustness.

**Acknowledgements**. This work was partially conducted while Nasim Rahaman was interning at Meta AI. Chris Pal and Martin Weiss thank NSERC for support under the COHESA program and IVADO for support under their AI, Biodiversity and Climate Change Thematic Program. Nasim Rahaman and Bernhard Schölkopf was supported by the German Federal Ministry of Education and Research (BMBF): Tübingen AI Center, FKZ: 01IS18039B, and by the Machine Learning Cluster of Excellence, EXC number 2064/1 - Project number 390727645.

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
