# A  Graph Priors

## A.1  Graph Structure Regularizer

In this section, we describe in detail the graph structure regularizer, as introduced in Section 2.2.

**Overview of the Objective.** The graph structure regularizer applies on the link probability matrix $P_{ij}$ introduced in Equation 2, and ensures that it matches a *canonical* link probability matrix (see below) of an arbitrary prior random graph $P_{ij}^{(0)}$, up to a permutation of the node labels. Where $\sigma_*$ is this permutation (see below), the overall regularization objective given by

$$\mathcal{L}_{\text{graph}} = \sum_{i \neq j} \left( P_{ij} - P_{\sigma_*(i)\sigma_*(j)}^{(0)} \right)^2 . \tag{6}$$

Here we use the mean-squared error, but other distance metrics (e.g. KL divergence) are possible. Further, we exclude the diagonals (where $i = j$) because $P_{ii} = 1$ by definition, since $d(\boldsymbol{s}_i, \boldsymbol{s}_i) = 0$ in Equation 2. Given this overall objective, we now discuss how we obtain its key components: the prior $P_{ij}^{(0)}$ and the permutation $\sigma_*$.

**Prior Link Probabilities.** Before defining the prior link probability matrix $P_{ij}^{(0)}$, we informally introduce the notion of graph functions, or *graphons* (a formal yet accessible treatment can be found in [19]). A graphon is a mathematical tool for describing certain classes of random graphs where the node labels are exchangeable, i.e. graphs where the permutation over nodes does not matter. It enables one to sample the probability that two nodes labeled $i$ and $j$ in a randomly sampled graph are connected, which in turn is used to sample whether or not there is an edge between the nodes $i$ and $j$ in the random graph.

Mathematically, a graphon $W : [0, 1] \times [0, 1] \to [0, 1]$ is a symmetric, Lebesgue measurable [8] function mapping from the unit square $[0, 1]^2$ to the unit interval $[0, 1]$. To sample a random graph with $U$ nodes, we first draw $U$ samples from the uniform distribution on the unit-interval $\mathcal{U}([0, 1])$, and call these $r_1, \cdots, r_u, \cdots, r_U$. Now, the probability that there is an edge between the nodes $i$ and $j$ in the sampled random graph is given by $W(r_i, r_j)$. We observe that $r_u$ being sampled (for all $u$) establishes the exchangeability of nodes.

To define the prior link probability matrix $P_{ij}^{(0)}$, we select some canonical ordering of nodes (more on this below), e.g. $r_u = {}^u\!/_{U-1}$ where $u \in \{0, \cdots, U - 1\}$, and define:

$$P_{ij}^{(0)} = W(r_i, r_j) \tag{7}$$

In words, $P_{ij}^{(0)}$ is simply the graphon $W$ sampled on a grid.

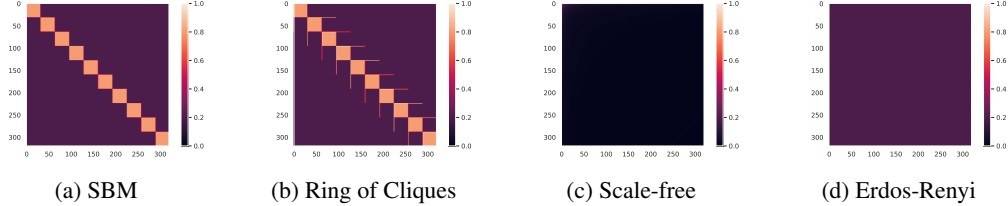

| (a) SBM | (b) Ring of Cliques | (c) Scale-free | (d) Erdos-Renyi |

Figure 7: **Sampled Graphons.** The quantity $P_{ij}^{(0)}$ for the graphs considered in this work, for $U = 320$.

As to the choice of a graphon $W$, we consider several options, including stochastic block models (SBMs), ring-of-cliques, scale-free model, and the Erdös-Renyi model – Figure 7 shows the corresponding sampled graphons. The simplest of these corresponds to the Erdös-Renyi model [18], where all pairs of nodes are equally likely to be connected, in which case $W(r_i, r_j) = $ constant. For the scale-free network, we use the following graphon with parameter $\beta$ (which we set to 0.5):

$$W(r_i, r_j) := \frac{U^\beta}{16}(r_i + 1)^{-\beta}(r_j + 1)^{-\beta} \tag{8}$$

**Permutation over Graph Nodes.** Recall that we obtained $P_{ij}^{(0)}$ by ordering the nodes in a canonical but arbitrary manner. However, permuting the node labels does not change the relevant properties of the underlying graph. To account for this, we allow for arbitrary permutations of node labels in Equation 6, meaning that the modules (labeled by indices $i$ and $j$ in $P_{ij}$ in Equation 6) must not adhere to the canonical order defined for $P_{ij}^{(0)}$. However, this exposes us to a difficult challenge of inferring the correct permutation over node labels (recall that there are $U!$ such permutations where $U \approx \mathcal{O}(1000)$). We approach this heuristically by leveraging the Hungarian method to obtain an approximate solution to the search over all $U!$ possible permutations.

To this end, let $\sigma : \{0, \cdots, U-1\} \to \{0, \cdots, U-1\}$ be some permutation (i.e. a bijection) over the set of node labels. Let $C$ be a cost matrix, whose entries $C_{vw}$ measure the cost associated with mapping $\sigma(v) \mapsto w$. We define this as:

$$C_{vw} = \sum_i \left( P_{vi} - P_{wi}^{(0)} \right)^2 \tag{9}$$

The cost matrix is fed to a linear assignment problem solver (e.g. Hungarian method, accesible via `scipy.optimize.linear_sum_assignment`) to obtain the permutation $\sigma_*$. An implementation detail is that this solver is supported only on CPUs, necessitating a device transfer to move $C_{vw}$ to the CPU. However, in the unconditional setting, this matching must happen only once per iteration (and not once per sample in the batch, like in e.g. DETR [13]); consequently, the overhead is not significant and we observe good GPU utilization.

## A.2 Illustration of Learned Graphs

We visualize the learned graphs in Figure 8.

# B ModFFN and SKMDPA

## B.1 ModFFN Parameter Example

The majority of ModFFN parameters are shared between modules, but a small number of parameters (the code vectors, in the unconditional case) are not. Let's consider a concrete example — a 2 layer deep ModFFN that takes in $d_{in} = 384$ dimensional inputs and returns $d_{out} = 384$ dimensional outputs, with a hidden layer size of $d_h = 1536$. Let us assume that the number of modules is $U$, each associated with a code vector of dimension $d_c = 384$. Let us further assume that the total number of such ModFFNs in the network is $L = 8$. The total number of parameters in all ModFFNs is given by: $L \cdot (2d_{in} \cdot d_h + 2 \cdot d_c \cdot d_h) + U \cdot d_c = 1.89 \times 10^7 + 384 \cdot U$. The multiplier 2 is due to the fact that each FFN has two layers. We can observe that the number of parameters increases only very modestly with the number of modules $U$. It would take more than 10000 modules before the contribution due to code vectors starts becoming noticeable.

## B.2 SKMDPA

The notion of a module in NACs generalizes that of a latent vector (a row in the latent array) defined in Perceiver IO [28]. Recall from Section 2 (Circuit Design) that each module in NACs can be described by three vectors: a signature vector, a code vector, and an initial state. The initial state exactly corresponds to a latent vector in Perceiver IO, and therefore, the number of modules exactly equals the number of latent vectors in Perceiver IO. However, we note that there is no equivalent of signature and code vectors in Perceiver IO.

In this context, SKMDPA in NACs is functionally a drop-in replacement for the transformer-based attention in Perceiver IO. Like transformer-based attention, it also consumes $U$ input vectors and outputs $U$ output vectors. Unlike transformer-based attention where all input vectors can influence all output vectors, only some input vectors can influence some other output vectors in SKMDPA. Which vector influences which other vector is specified by the corresponding signatures (recall that there are exactly $U$ of these), which are also learned.

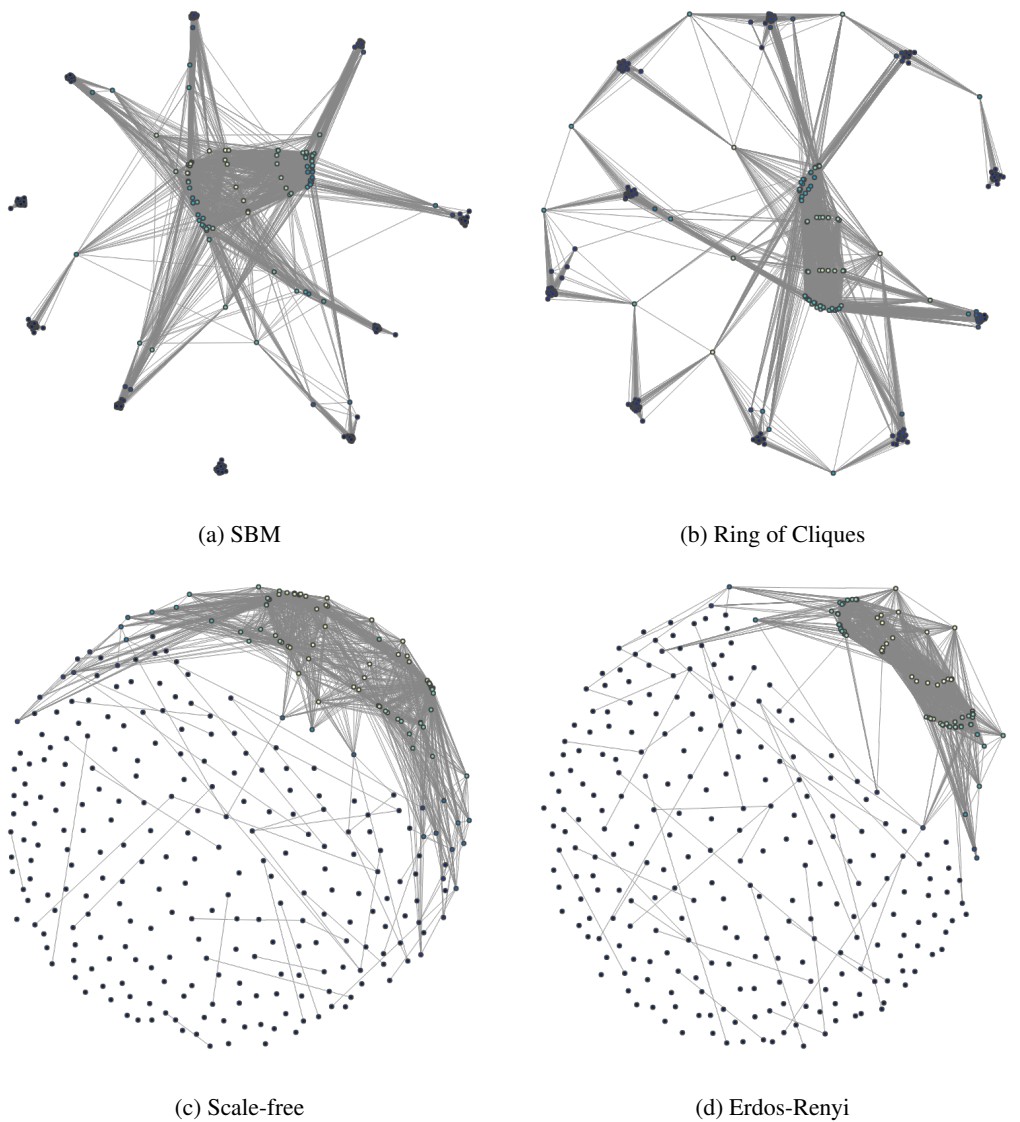

(a) SBM

(b) Ring of Cliques

(c) Scale-free

(d) Erdos-Renyi

Figure 8: **Visualization of the Learned Graphs.** We draw a an edge between two nodes $i$ and $j$ if the corresponding probability $P_{ij}$ (see Equation 2) is larger than $0.5$. Further, the distances between any pair of nodes in the visualized graphs are optimized to be proportional to the distance between their corresponding signatures.

# C Pre-training Details

**Data Augmentation.** We used a standard data augmentation regimen [50, 33] when pretraining on (Tiny-)ImageNet for both NACs and PIOs.

**Convolutional Preprocessor.** For Tiny-ImageNet, the image tokenization pipeline was the same across both models. For each, we downsample twice: first by a factor of 4 (via a convolutional layer with kernel size and stride of $4$, followed by a layernorm), and then by another factor of two (via another convolutional layer with kernel size and stride of $2$, followed by a layernorm). Between the two downsampling blocks, we include a ConvNeXt [35] block with a single convolutional layer (with a kernel size of 7 and no stride) followed by two point-wise linear layers. Each of the linear layers in the ConvNeXt block is preceded by a layernorm, and there is a GELU activation after the first linear layer. The output of the preprocessor is a feature-map of size $8 \times 8$ for Tiny-ImageNet and $28 \times 28$ for ImageNet. We concatenate a positional encoding to this feature-map before feeding it to the model. These encodings are learnable vectors, each with 64 dimensions. **Convolutional Preprocessor.** For Tiny-ImageNet, the image tokenization pipeline was the same across both models. For each, we downsample twice: first by a factor of 4 (via a convolutional layer with kernel size and stride of $4$, followed by a layernorm), and then by another factor of two (via another convolutional layer with kernel size and stride of $2$, followed by a layernorm). Between the two downsampling blocks, we include a ConvNeXt [35] block with a single convolutional layer (with a kernel size of 7 and no stride) followed by two point-wise linear layers. Each of the linear layers in the ConvNeXt block is preceeded by a layernorm, and there is a GELU activation after the first linear layer. The output of the preprocessor is a feature-map of size $8 \times 8$ for Tiny-ImageNet and $28 \times 28$ for ImageNet. We concatenate a positional encoding to this feature-map before feeding it to the model. These encodings are learnable vectors, each with 64 dimensions.

**Hyperparameters for Tiny-ImageNet.** Table 4 shows all hyperparameters we used to train NACs and PIOs on Tiny-ImageNet. For Perceiver IO, we adapt a popular third-party implementation[1]. The learning rate and weight decay were tuned individually for each model. We also note that the weight decay is neither applied to parameters with dimension less than or equal to 1, nor codes and signatures.

**Hyperparameters for ImageNet.** Table 5 shows all hyperparameters we used to train NACs on ImageNet. We use a pretrained PIO model produced by [28] and made available in Huggingface's transformers repository[2].

# D Limitations & Future Work

**Future work.** Modular, general-purpose neural architectures present many opportunities for future work. These include (1) incorporating train-time sparsity where each input sample or token only updates a small fraction of the modules for one training step, (2) adding sparse kernels to the implementation to support sparsification, (3) conditioning the configuration of modules based on the task (perhaps in addition to sample-conditional generation), (4) thorough analysis of scaling laws associated with such architectures, (5) development of a principled approach to performing sparse updates to modules, (6) learn a predictive model that identifies a subnetwork of modules for conditional computation, and (7) train modules to solve certain hard-coded problems and learn how to wire them.

**Limitations.** In addition, this work has several limitations. First, pre-training of NAC on large-scale natural language datasets (such as C4 [43]) and on large-scale multi-modal datasets was determined to be out of scope. Future work may build on this architecture and determine the best way to perform this large-scale pre-training, and the incorporation of self-supervised objectives. Second, while our approach for learning the circuit generator via signatures and a learned cross attention mechanism was sufficient for validating that NACs can learn sample-conditional module configurations, we expect that further experimentation with auto-regressive models and GFLowNets [12]) will yield better configurations. Third, an analysis of the scaling properties of NACs as compared to homogenous general-purpose modules would likely be useful for those seeking to compare against this model-class with access to different amounts of compute resources.

---

[1]`https://github.com/lucidrains/perceiver-pytorch`
[2]`https://huggingface.co/docs/transformers/model_doc/perceiver`

Table 4: Hyperparameters for training on Tiny-ImageNet.

| Hyperparameter | Neural Attentive Circuit | Perciever IO |
| --- | --- | --- |
| Batch size | 1024 | 1024 |
| Number of epochs | 400 | 400 |
| Augmentation pipeline | Standard [33, 50] | Standard [33, 50] |
| Weight decay | 0.05 | 0.05 |
| Optimizer | AdamW | AdamW |
| Learning rate scheduler | Cosine | Cosine |
| Base peak learning rate (for batch size 512) | 0.0003 | 0.0005 |
| Base min learning rate (for batch size 512) | 1e-5 | 1e-5 |
| Warmup epochs | 25 | 25 |
| Warmup from learning rate | 1e-6 | 1e-6 |
| Dimension of state ($d_{model}$) | 384 | 384 |
| Layers ($L$) | 8 | 8 |
| Layers that do not share weights | 8 | 8 |
| Processor modules (NACs) or latents (PIOs) ($U_p$) | 320 | 320 |
| Attention heads | 6 | 6 |
| Read-in (NACs) or cross-attention (PIOs) heads | 1 | 1 |
| Activation function | GEGLU | GEGLU |
| FFN hidden units | 1536 | 1536 |
| Output modules ($U_o$) | 64 | N/A |
| Signature dimension ($d_{sig}$) | 64 | N/A |
| Code dimension ($d_{code}$) | 384 | N/A |
| Sampling temperature ($\tau$) | 0.5 | N/A |
| Kernel bandwidth ($\epsilon$) | 1.0 | N/A |
| Modulation $\alpha$ at initialization | 0.1 | N/A |

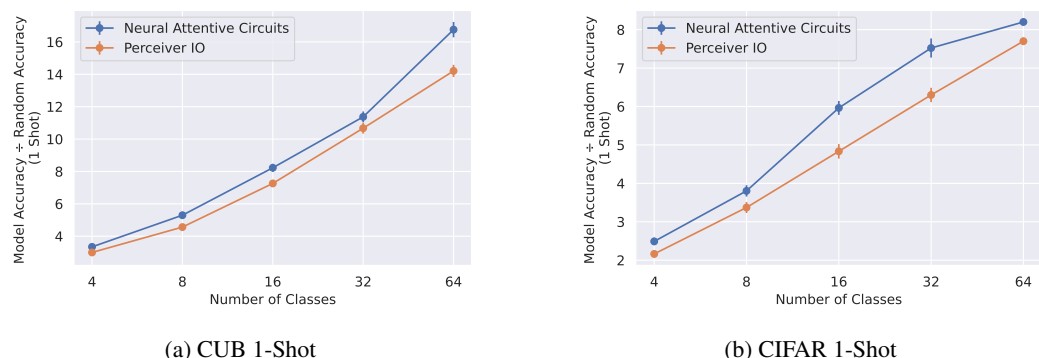

(a) CUB 1-Shot  (b) CIFAR 1-Shot

Figure 9: We evaluate NACs and the baseline PIO on CUB (left) and CIFAR (right) while varying the number of classes between 1 and 64 (x-axis) and holding the number shots constant at 8. On the Y-axis we show the lift, which is the model accuracy divided by the random accuracy for the task. We see that the NAC remains competitive across the board with respect to 1-shot performance.

## E  Additional Fewshot Experiments

Figure 9 shows the result of varying the number of ways in the few-shot task. This experiment demonstrates that NACs maintain a consistent improvement over the PIO baseline across a wide range of task difficulties.

## F  Computational Complexity Comparison

**Selecting what modules are dropped.** After pre-training on Tiny-ImageNet, we construct a measure of importance of a module. To this end, we consider the link probability matrix $P$ (see Equation 2),

Table 5: Hyperparameters for training on ImageNet. Note that we do not train the Perceiver IO, but use a pre-trained model. Nevertheless, we gather available hyperparameters from Jaegle et al. [28] and the official code-release for the reader's convenience.

| Hyperparameter | Neural Attentive Circuit | Perciever IO |
|---|---|---|
| Batch size | 1024 | 1024 |
| Number of epochs | 110 | 110 |
| CutMix | ✓ | ✓ |
| MixUp | ✓ | ✓ |
| RandAugment | ✓ | ✓ |
| Augmentation pipeline | Standard [33, 50] | Custom [28] |
| Weight decay | 0.05 | 0.1 |
| Optimizer | AdamW | LAMB |
| Learning rate scheduler | Cosine | Custom[28] |
| Base peak learning rate (for batch size 512) | 0.0003 | 0.001 |
| Base min learning rate (for batch size 512) | 1e-5 | 0 |
| Warmup epochs | 25 | N/A |
| Warmup from learning rate | 1e-6 | N/A |
| Dimension of state ($d_{model}$) | 512 | 1024 |
| Layers ($L$) | 8 | 48 |
| Layers that do not share weights | 8 | 6 |
| Processor modules (NACs) or latents (PIOs) ($U_p$) | 960 | 512 |
| Attention heads | 8 | 8 |
| Read-in (NACs) or cross-attention (PIOs) heads | 1 | 1 |
| Activation function | GEGLU | GELU |
| FFN hidden units | 1024 | 1024 |
| Output modules ($U_o$) | 64 | N/A |
| Signature dimension ($d_{sig}$) | 64 | N/A |
| Code dimension ($d_{code}$) | 512 | N/A |
| Sampling temperature ($\tau$) | 0.5 | N/A |
| Kernel bandwidth ($\epsilon$) | 1.0 | N/A |
| Modulation $\alpha$ at initialization | 0.1 | N/A |

where $P_{ij}$ gives a measure of how strongly module $i$ is connected with module $j$. To obtain the importance measure of the module at index $i$, we define the quantity $q_i$ as following:

$$q_i = \sum_j P_{ij} \tag{10}$$

We then sort the modules by their respective $q_i$, and drop the ones for which $q_i$ is smallest. This ensures that the most strongly connected modules are dropped last.

**Eliminating connections between modules.** In addition to dropping modules, we also investigate the effect of prohibiting attentive connections between modules (Figure 10). To this end, we sort $P_{ij}$ and successively eliminate the smallest elements by setting them to 0. For instance, if 90% of connections are removed, we have that 90% of elements in the lower and upper triangular portion of the matrix $P_{ij}$ (excluding the diagonal) are set to 0.

From Figure 10, we make two **key observations: (a)** A large fraction ($\approx 90\%$) of connections can be disallowed without incurring a significant loss in performance. In this regime, the SBM and Ring-of-Cliques priors (with 59.46% and 59.91% validation accuracy, respectively) are advantaged over the scale-free prior (with 59.1% validation accuracy), even though the latter obtains the best validation performance given the full model (60.76%). **(b)** As more edges are disallowed, we find that the SBM and Ring-of-Cliques prior abruptly break down. However, the scale-free and Erdös-Renyi priors keep gracefully degrading until eventually almost all connections are prohibited.

Further, we investigate how the graph over module changes as connections are dropped, e.g. around the critical value for SBM and Ring-of-Cliques priors. We discover interesting patterns, as shown in Figures 11, 12, 13 and 14.

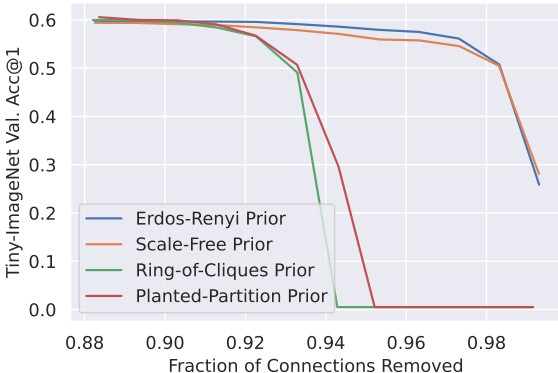

Figure 10: **Effect of eliminating connection between modules.** We find that a relatively large fraction ($\approx 90\%$) of module connections can be disallowed while only marginally affecting performance. Beyond that, we see a stark difference in behaviour: on the one hand, for ring-of-clique and stochastic block model priors, we find that beyond a critical amount of removed connections, the performance degrades drastically. On the other hand, we find that scale-free and Erdös-Renyi priors degrade gracefully as eventually all connections are disallowed.

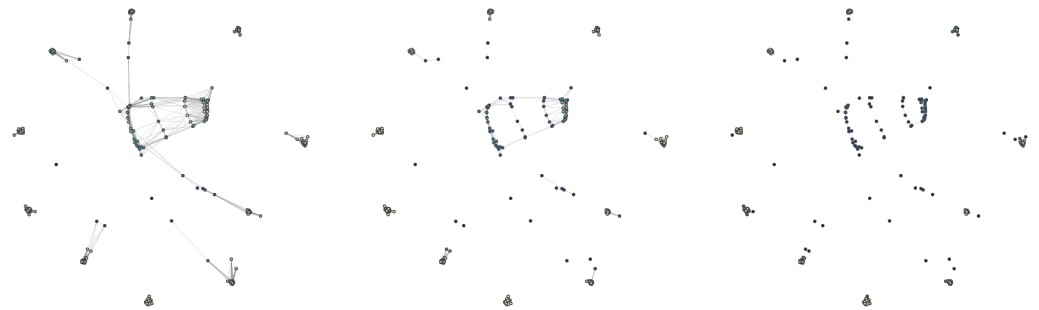

(a) 50.69% acc. @ 93% edges lost.    (b) 29.62% acc. @ 94% edges lost.    (c) 05.00% acc. @ 95% edges lost.

Figure 11: **Eliminating attentive connections (edges) in a NAC trained with SBM Prior.** We observe that the accuracy drops drastically as the edges relaying information between clusters of modules are eliminated. Further, we see the emergence of four core clusters over modules, and the performance is essentially random when the attentive connections between these clusters are severed.

Finally, while we leave a thorough benchmarking to future work, we note that this behaviour allows for the use of sparse attention kernels. Even without invoking sparse attention kernels, it is possible to extract an inference-time speed-up by leveraging the block-sparse structure of the attention sparsity mask, for e.g. the SBM prior.

**Details around inference speed measurements.** For Figure 5, we measured the time per sample for the NAC on an A100 GPU. We used a batch-size of 64 and timed the model under a `torch.inference_mode` context manager.

**Clarifying the effect of sparsification**. Figure 15 clarifies the effect of sparsification on NACs (with different graph priors) and Perceiver IO. The figure shows GFLOPs / sample on the X-axis and Tiny-ImageNet val accuracy on the Y-axis.

We vary the GFLOPs by dropping modules in NACs or latents in Perceiver IO. We make two observations: for NACs, even when the FLOP budget is reduced by a factor of 5 (to roughly 1.6 GFLOPs), the amount of performance lost is remarkably small, corresponding to e.g., roughly 1% in accuracy difference with respect to the full model with ring-of-cliques prior. In contrast, Perceiver IO loses 5.3% in validation accuracy with respect to the full model when given roughly the same amount of FLOPs (1.7 GFLOPs). Under 1 GFLOP, Perceiver IO degrades slower than NACs. This is because

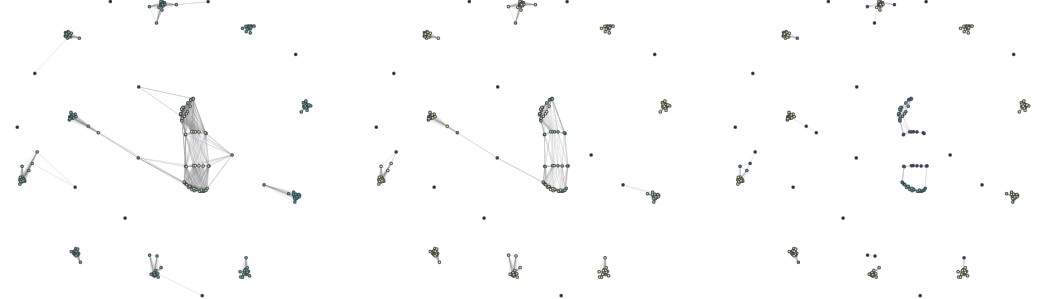

(a) 56.59% acc. @ 92% edges lost.  (b) 49.09% acc. @ 93% edges lost.  (c) 05.00% acc. @ 94% edges lost.

Figure 12: **Eliminating attentive connections (edges) in a NAC trained with Ring-of-Cliques Prior.** Like in Figure 11, observe the emergence of core clusters, but also important relay modules that connect multiple clusters.

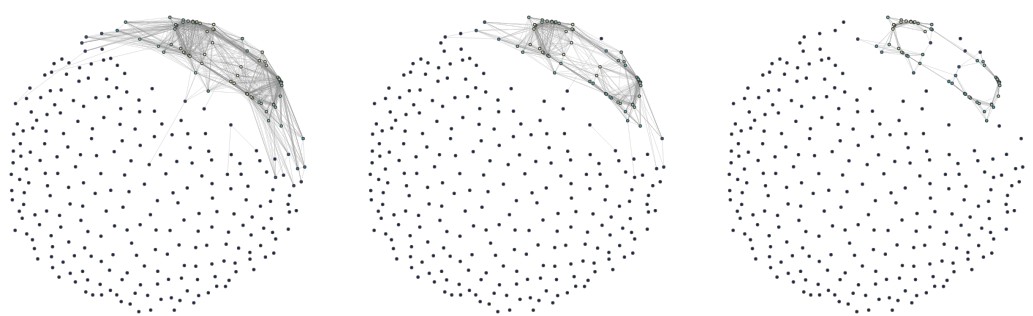

(a) 54.56% acc. @ 97% edges lost.  (b) 50.51% acc. @ 98% edges lost.  (c) 28.08% acc. @ 99% edges lost.

Figure 13: **Eliminating attentive connections (edges) in a NAC trained with Scale-free Prior.** We observe the emergence of a cluster of well-connected *hub* modules, together with a tail of less connected modules. Severing the connections to the tail modules while just keeping the hub modules connected results in a roughly 10% decline in validation accuracy. But as the connections between the hub modules are severed, the performance significantly drops.

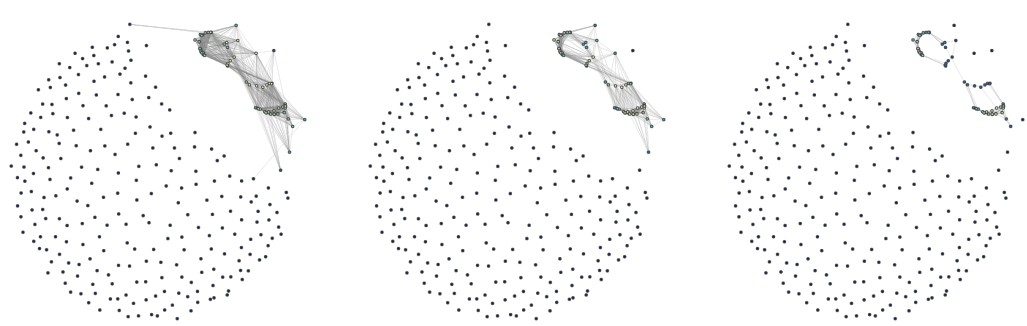

(a) 56.13% acc. @ 97% edges lost.  (b) 50.75% acc. @ 98% edges lost.  (c) 25.88% acc. @ 99% edges lost.

Figure 14: **Eliminating attentive connections (edges) in a NAC trained with Erdös-Renyi Prior.** We see a pattern similar to the one in Figure 13. Interestingly, we observe that while the Erdös-Renyi prior encourages uniform connectivity between nodes (i.e. it is an uninformative prior), the learned graph resembles a scale-free network, which happens to perform best in the IID regime.

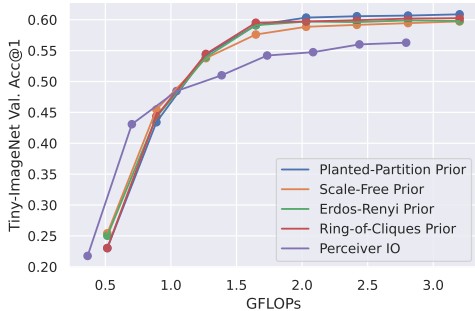

Figure 15: **Effect of Sparsification on NACs and Perceiver IO.** We show the effect of dropping modules (in NAC) and latents (in Perceiver IO) on Tiny-ImageNet validation accuracy and compute.

the cross attention in Perceiver IO is computationally lighter than the read-in attention in NACs. This difference only becomes visible in the very low compute regime (i.e., under 1 GFLOP), where the read-in attention's fixed compute cost is not outweighed by the other attention mechanisms and FFNs. We further note that the read-in attention in NACs can be replaced by the computationally cheaper cross attention like in Perceiver IO, at the cost of a modest amount of performance (corresponding to roughly 1% validation accuracy on Tiny-ImageNet).

## G   NLVR2 Experiments

We begin by evaluating the zero-shot performance of OpenAI's CLIP model on NLVR2. CLIP takes in a collection of images and captions and produces a similarity score between the two modalities. In order to get CLIP to produce logits, we initially attempted to mimic the evaluation setup of the NLVR2 leaderboard by providing CLIP with a concatenated pair of input images and two sentences. The first sentence was provided in the NLVR2 dataset, while the second was generated by prompt engineering GPT-J-6B to produce a negated sentence. We **achieved a 53% zero-shot accuracy** on this task, only 3% above the random chance baseline. For context, the official NLVR2 leaderboard, would place zero-shot CLIP in fourth place just behind VisualBERT which achieved a 67% accuracy, and above methods such as MaxEnt (54%), FiLM[40] (51%), N2NMN [26] (51%) and MAC Networks [27] (50.8%).

In this set of experiments, we use CLIP as a frozen backbone network and apply the image and text encoder to each modality. We take then take the output encoded tokens and fine-tune a Perceiver IO and a Neural Attentive Circuit on these, following the typical NLVR leaderboard evaluation. With Perceiver IO, we see an improvement in the **validation accuracy to 64%**.

**Conditional NAC Architectural Details**. In our experiments with conditional NACs, the circuit generator was implemented by a simple cross attention layer followed by a feed forward network (FFN), and a self-attention layer followed by two FFNs run in parallel (more on that below). In the initial cross-attention layer, the keys and values were derived by linearly transforming pre-pooling CLIP text embeddings (which yields one embedding vector per text token). We used CLIP-base (the smallest available pre-trained CLIP) to minimize the compute requirement. The queries in the first cross-attention layer were learned 384-dimensional vectors, of which we had exactly $U$ (i.e. one per module). The output of the self attention layer was therefore another set of $U$ vectors, each of which was passed through two different FFNs in parallel. The first FFN produced the signatures for each module ($U$ 64-dimensional vectors), whereas the second FFN produced the corresponding codes ($U$ 384-dimensional vectors).

## H   Text Embedding Experiments

**Text Dataset Construction.** In order to evaluate the text embeddings produced by the NAC circuit generator, we constructed a small dataset of simple statements. These statements involve different semantics including *hard cardinality* which includes a precise number of objects, *soft cardinality*

which includes words like "many" or "few", *existential* which only indicates that an object is present, and *spatial relationships* which use a prepositional phrase to denote a physical relationship. We constructed a simple template for each reasoning skill,

1. **hard cardinality**: "There are exactly {integer} {object(s)}".
2. **soft cardinality**: "There are many {object(s)}".
3. **existential (singular)**: "There is a {object}".
4. **existential (plural)**: "There are any {objects}".
5. **existential (adjective)**: "There is a {color} {object}".
6. **spatial relationship**: "There is a {object} {spatial relationship} {object}".

and constructed a statement for each combination of our chosen set of integers, colors, objects (we combined animals into the objects category), and spatial relationships. The sets we selected are as follows:

1. **integers**: "two", "three", "four", "five", "six", "2", "3", "4", "5", "6"
2. **animals**: "cat", "dog", "mouse", "cheetah", "stingray", "jellyfish", "guinea pig", "marmot", "chimp", "gorilla"
3. **objects**: "boat", "flute", "door", "laptop", "ball", "machine", "paper towel"
4. **colors**: "blue", "black", "red", "green", "yellow", "purple", "orange", "white", "teal", "beige", "pink", "gray", "pink"
5. **spatial relationships**: "near", "far away from", "next to", "on top of", "under", "behind", "on", "by", "above", "in front of"

**CLIP Text Embeddings**. We evaluate the CLIP text embeddings using the same text dataset used to assess the conditional NAC's link probability matrix generation in Figure 6. We see in Figure 16 that CLIP's latent space is primarily organized around semantic categories such as "ball" or "dog". This stands in contrast with NAC's embeddings which are organized by abstract reasoning tasks such as counting (hard cardinality) and spatial relationships. Figure 17 shows the same TSNE embedding but points to the same samples as referenced in the main paper in Figure 6.

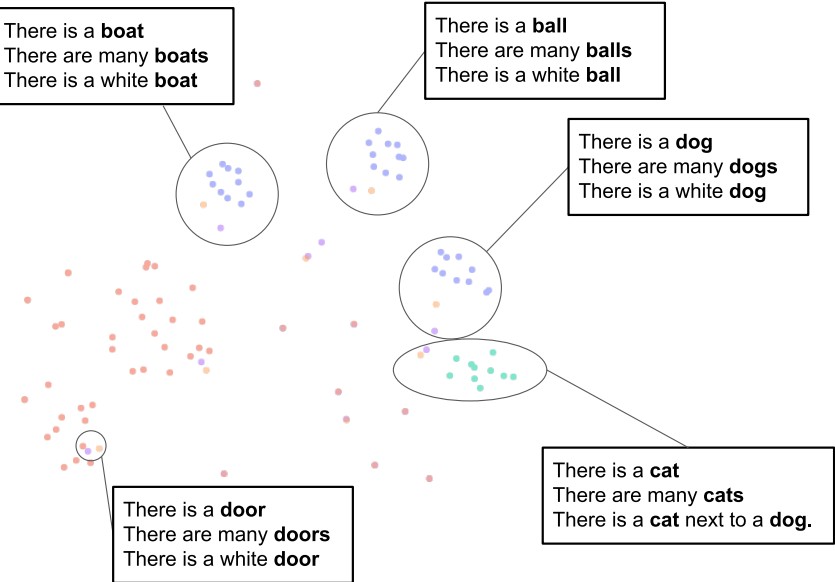

Figure 16: **CLIP Text Embeddings**. This plot shows a TSNE of CLIP's pooled output for the same sentences. We can see that CLIP's latent space is primarily organized by semantic category, not cardinality, spatial relationships, or quantification.

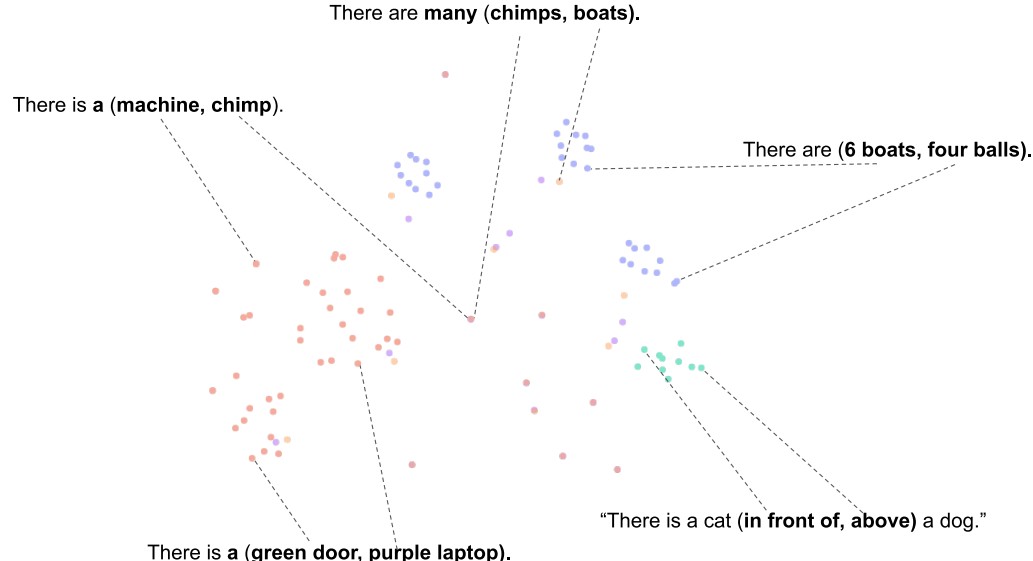

Figure 17: **CLIP Text Embeddings**. This plot shows a TSNE of CLIP's pooled output for the same sentences. We can see that CLIP's latent space is primarily organized by semantic category, not cardinality, spatial relationships, or quantification.

# I  Anonymized Code

Anonymized research code can be found here[3].

---