# OpenReview forum: "Neural Attentive Circuits"
_NeurIPS.cc/2022/Conference — NeurIPS 2022 Accept_

### Official Review · Reviewer_p9jH · 2022-07-11

**Rating:** 6
**Confidence:** 3
**Soundness:** 3 good
**Presentation:** 3 good
**Contribution:** 2 fair

**Summary:**

The paper presents a novel neural architecture (NAC) which learns to jointly predict the optimal (sparse) connectivity (via signatures) and parameterization (via codes) of its general purpose modules (ModFC/FFN). The architecture appears parameter- and computation-efficient, compares favorably against Perceiver IO in terms of few-shot adaptation (CUB/CIFAR) and OOD generalization (Tiny-ImageNet), and meaningfully generates similar connectivity patterns when conditioned on similar input samples (NLVR2).

**Questions:**

- All experiments seem to use some form of preprocessing (convolutional or CLIP) which in a way renders the general purpose claim of the paper questionable. Does the proposed architecture not work with raw inputs (i.e. byte arrays) like Perceiver IO?
- In Table 1, is (Perceiver IO + ModFC) equivalent to (NAC - sparse graph prior) (i.e. without minimizing Eq 9)? Shouldn’t a dense connectivity perform better under ID validation due to less constraints? Mechanistically, how does the spare connectivity in NACs help achieve better OOD generalization?


**Limitations:**

The authors have adequately addressed the limitations of their work.

**Strengths And Weaknesses:**

Strengths
+ (Originality, Major) The paper’s novel architectural features (ModFC, sparse graph priors) compared to Perceiver IO, and performance improvements (linear scaling w.r.t. input size, adaptive computation with large speed boost) compared to Neural Interpreters, are impressive.
+ (Clarity) The writing is clear and easy to follow. The figures are well-made and very helpful.

Weaknesses
- (Quality) The empirical validation of the proposed architecture is insufficient in my opinion. To claim that NAC is a stronger general purpose architecture, the scope of the comparisons (now only few-shot adaptation and OOD generalization in CV datasets) and the number of baselines (now only Perceiver IO) both need to be reasonably improved. E.g. Perceiver IO was evaluated on 6 different modalities (Table 5 in their paper) and against non-general purpose baselines within each modality.
- (Significance)
Due to the insufficient empirical validation, it’s hard to conclude that the proposed architecture is significant and will inspire or lead to future breakthroughs.

---

> ### Author Response · Authors · 2022-08-01
> **Author Response to Reviewer p9jH [Part 1]**
>
> Thank you for taking the time to review our paper and provide helpful feedback. We have added several empirical evaluations that demonstrate NACs are a useful modularization of general purpose models. We believe that these additional experiments meaningfully improve the significance and quality of the work, and that your feedback encouraged us to perform them.
>
> ### Empirical results that demonstrate NACs are a General Purpose architecture
>
> We have broadened the scope of our experiments section to address the primary weaknesses as assessed by your review. Our new experiments span diverse data modalities (see below) while outperforming full-attention transformers and linear attention Linformers.
>
> The new modalities we investigate in addition to our original experiments on images and images + text are (a) ModelNet40 point clouds, (b) ListOps symbolic processing, and (c) IMDB text classification from a raw byte-stream (i.e., without text tokenization). Where applicable, we cite where the results are collected from the literature or specify that we performed the evaluations.
>
> **Point clouds.** A DeepMind team trained the Perceiver family of models on the point cloud classification benchmark ModelNet40 without additional data augmentation, geometric inductive biases or self-supervised pre-training [6]. In that setting, Perceiver IO obtained 77.4%, Hierarchical Perceivers achieved 75.9% without self-supervised pre-training, and when trained with self-supervised pre-training Hierarchical Perceivers obtained 80.6% (cf. Table 7 of [6]). We ran a comparable NAC without specialized tokenization or pre-training and achieved an 83% test accuracy. It appears that the sparse inductive prior was quite effective in this setting.
>
> | Method | Test Accuracy |
> | --- | --- |
> | Hierarchical Perceiver (No MAE) [6] | 75.9% |
> | Perceiver IO [6] | 77.4% |
> | Hierarchical Perceiver (With MAE) [6] | 80.6% |
> | **NAC (ours)** | **83.0%** |
>
> [6] Hierarchical Perceivers. [https://arxiv.org/abs/2202.10890](https://arxiv.org/abs/2202.10890)
>
> **ListOps Symbolic Processing.** The ListOps symbolic processing task is a challenging 10-way classification task on sequences of up to 2,000 characters. This commonly used benchmark [7] tests the ability of models to reason hierarchically while handling long contexts. The results are as follows:
>
> | Method | Test Accuracy |
> | --- | --- |
> | Full-Attention Transformers [7] | 37.13% |
> | Linformer [7] | 37.38% |
> | Perceiver IO (our impl.) | 39.70% |
> | **NAC (ours)** | **41.40%** |
>
> [7] Long Range Arena. [https://github.com/google-research/long-range-arena](https://github.com/google-research/long-range-arena)
>
> **Text Classification from Raw Bytes.** The IMDB task is a binary classification of sentiment on sequences of up to 4,000 bytes. In particular, we note that the models ingest raw ascii bytes, and **not** tokenized text.  All reported methods are trained without pre-training or data augmentation. The results are as follows:
>
> | Method | Test Accuracy |
> | --- | --- |
> | Full-Attention Transformers [7] | 65.35% |
> | Linformer [7] | 56.12% |
> | Perceiver IO (our impl.) | 66.50% |
> | **NAC (ours)** | **68.18%** |
>
> In addition to these experiments, we remark that the ImageNet few-shot results reported in Figure 4 also compares against a pre-trained Perceiver IO that has been trained by Jaegle et al. 2021 [8], the authors of Perceiver IO.
>
> [8] Perceiver IO. [https://arxiv.org/abs/2107.14795](https://arxiv.org/abs/2107.14795)
>
> ---
> **Continued in Part 2**

---

> > ### Author Response · Authors · 2022-08-01
> > **Author Response to Reviewer p9jH [Part 2]**
> >
> > ### How does Sparse Connectivity Help?
> >
> > > In Table 1, is (Perceiver IO + ModFC) equivalent to (NAC - sparse graph prior) (i.e., without minimizing Eq 9)? Shouldn’t a dense connectivity perform better under ID validation due to less constraints? Mechanistically, how does the spare connectivity in NACs help achieve better OOD generalization?
> > >
> >
> > Your interpretation of the methods is indeed correct: (Perceiver IO + ModFC) is equivalent to (NAC - sparse graph prior), i.e. with a complete (all-to-all) graph.
> >
> > We appreciated your question on why learnable sparsity improves IID performance over dense connectivity. We believe that the answer rests on a central insight about the importance of integrating modularity in general-purpose neural architectures, and is a core contribution of this work. Specifically, if we believe that the true underlying data generating process can be expressed as a system of [sparsely interacting independent mechanisms](https://arxiv.org/pdf/2102.11107.pdf), then we should prefer models that are learnably sparse.  If we enforce hard (unlearnable) sparsity, the model is too rigid (underfitting) and fails to capture important aspects of the data generative process (cf. third entry in Table 1). But if we do not enforce any sparsity, the model is too flexible (overfitting) and therefore learns solutions that fail to generalize better (cf. second entry in Table 1). We hypothesise that NACs live in a *[Goldilocks zone](https://en.wikipedia.org/wiki/Goldilocks_principle)* in terms of model complexity.
> >
> > A similar argument can be made for OOD performance. Under the assumption of sparsely interacting independent mechanisms, natural distribution shifts tend to arise from a sparse and localised shifts in mechanisms. If NACs better capture this sparse structure, we might expect such natural distribution shifts to result in sparse disruptions in the model; i.e., we might expect that some NAC modules might still function nominally, and that disruptions are localized. If we instead have a densely connected model operating under a sparse distribution shift, then the model may be more globally disrupted.
> >
> > To conclude, we thank you again for the effort you have invested in reviewing our work. Please let us know if you have further questions.

---

> ### Author Response · Authors · 2022-08-06
> **Thank you again for your review.**
>
> We thank you again for your constructive review. We were happy to learn that you found our work’s originality as a major strength, and our writing clear and easy to follow.  Your primary concern was about the strength of the empirical results that demonstrate the general purpose nature of NACs. Based on this feedback, we have run additional experiments on three diverse domains. We found NACs to excel in each of these, also when compared against existing baselines in the literature (including vanilla transformers, linear attention transformers, and existing general purpose architectures).
>
> If these new results have changed your opinion of our work, we would appreciate it if you could consider updating your score. If not, please feel invited to continue interacting with us. Thank you.

---

> ### Comment · Reviewer_p9jH · 2022-08-07
> **Re: Author Response**
>
> Thank you very much for the additional results and explanation. I'm happy to increase my rating of the paper to 6. Below are some more suggestions and questions.
>
> 1. The additional results with more baselines are a positive change. However, more direct comparisons against the (near) SOTA baselines will be more ideal. E.g. PIO was compared against PWCNet and RAFT (near SOTA baselines in optical flow estimation) and performed [very competitively](https://paperswithcode.com/paper/perceiver-io-a-general-architecture-for) (#1 on KITTI, #2 & #5 on Sintel). Similarly, NAC actually performed [quite well on ListOps](https://paperswithcode.com/sota/long-range-modeling-on-lra) (#4) but was not mentioned.
> 2. IIUC, the original Perceiver actually got 85.7% on ModelNet40, considerably stronger than PIO's 77.4%. Does this mean PIO may not always be the right model to begin with, and maybe the original Perceiver + ModFC + sparse graph prior should be considered too?
> 3. The explanation for the spare connectivity using Table 1 is not really too convincing, mainly because the numbers are all very close with no standard deviation (or statistical significance) reported. In addition, the authors should consider more direct ways to validate their assumption about NAC's stronger capability in handling the naturally sparse data generation process, such as a) benchmarking NACs with different sparsity levels against synthetic datasets e.g. [50] with different amount of complexities or distribution shifts, or b) showing NACs attend to more general (rather than spurious) features via attention maps or XAI tools.
>
> [50] A Fine-Grained Analysis on Distribution Shift, ICLR, 2022.

---

> > ### Author Response · Authors · 2022-08-08
> > **Thank you for the additional feedback.**
> >
> > We appreciate that you are willing to upgrade your score given the additional results, and have taken the time to provide additional feedback. We respond to each item below.
> >
> > 1. Thank you for pointing this out. Given that the optical flow literature heavily relies on simulated training data and subsequent sim-to-real transfer, it would indeed be interesting to evaluate NACs in this setting in future work. Finally, we agree that NACs and Perceiver IO are quite competitive on ListOps, ranking #4 and #5 respectively.
> > 2. The original Perceiver relies on additional data augmentation to obtain 85% on ModelNet40. The numbers we report for Perceiver IO, NACs and HiPs [6] are without these additional augmentations. We do not know how well the original Perceiver would perform without the additional augmentations, but expect it to be in the same ballpark as Perceiver IO.
> > 3. Thank you for this feedback and appreciate your suggestions for future work around benefits of sparse inductive bias. We reply to each direction below.
> >     1. In our experiments we observed negligible variation in the results between each method (an order of magnitude smaller than the difference between methods), so we omitted these variances. We can add them back if the reviewer feels this would add to the convincingness of the table.
> >     2. Thank you or referencing the paper, A Fine-Grained Analysis on Distribution Shift, ICLR, 2022. We agree that this approach is an interesting one to probe the relationship between NACs sparse inductive bias and the dataset. In this work, we focused more on developing a method which would serve as an improvement over general-purpose models that could be used on a diverse set of real-world benchmarks rather than synthetic tasks, but we agree that further experimentation on synthetic tasks could help to understand their performance.
> >     3. We also agree that there are many opportunities for further qualitative investigation of the benefits of sparse connectivity. For example, there are many open questions about the relationship between complex real-world tasks and the related sparse prior graphs, which we are excited to pursue in future work.

---

> > > ### Comment · Reviewer_p9jH · 2022-08-08
> > > **Re: Table 1**
> > >
> > > Yes, thank you! I think most readers would also appreciate seeing the standard deviations reported.

---

### Official Review · Reviewer_g8fa · 2022-07-11

**Rating:** 7
**Confidence:** 4
**Soundness:** 3 good
**Presentation:** 3 good
**Contribution:** 3 good

**Summary:**

This paper presents a novel general purpose modular neural architecture; Neural Attentive Circuits. In NAC, information is routed through between a set of modules that perform conditioned computations. Information routing and computation conditioning is specified by the circuit generator and information is processed by the circuit executor. The paper proposes different instantiations for the circuit generator depending on the use case. NAC is compared to PerceiverIO, a general purpose architecture used as a basis for developing NAC, on image classification, few-shot adaptation and OOD generalization. Although NAC does not reach PIO’s performance level on the ImageNet validation set, it performs better on OOD generalization and Few-shot adaptation. Furthermore, pruning several modules significantly decreases the model’s computational cost without a significant loss in accuracy. Ablations of the model highlight the importance of design choices such the conditioned computations, modularity and sparsity priors. Finally, the usefulness of data conditioning in the circuit generator is demonstrated on a visual reasoning task where similar instructions generated similar circuit connectivity patterns.

**Questions:**

The paper lacks in clarity. It would be beneficial to explain in more detail the model design and experiments following the issues mentioned in weaknesses.

Since signatures and codes are shared across layers, the connectivity and modulation are similar in across layers. It would be interesting to vary dynamic signatures and codes while maintaining the weights shared across layers. Was model design explored previously ? If so, how did it perform compared to the current design ?

**Limitations:**

The limitations are addressed in the supplementary material and no negative societal impact was discussed in the paper.

**Strengths And Weaknesses:**

Strengths

- The paper proposes a novel method for modularizing neural networks.
- This method is has a flexible and efficient architecture design; module connectivity and modulation can be reduce the computational cost in the model and could potentially be incorporated in any model to replace MLPs.
- The model’s signature initialization can be used to embed specific priors in the model.
- A large percentage of modules can be pruned without significantly decreasing performance.
- Module connectivity provides insight on how the model solves tasks.
- The architecture design improves learning in low data regimes, OOD robustness and few-shot adaptation on two benchmarks.

Weaknesses

- While the architecture is claimed to be general purpose (applicable to arbitrary data types), its usefulness is demonstrated only on visual data.
- As the authors mentioned in the limitations, the method is not applied to large scale data compared to Perceiver and PerceiverIO. Furthermore, performance on the ImageNet validation set is significantly lower compared to PIO.
- If I understood section 2 correctly, the state $\theta^{i}_u$ of a module is only used to compute the query in layer $l+1$. If this is the case, the state does not represent a state of the model (as much as the weights $W$ and $W_c$). It would be less confusing to give $\theta^{i}_u$ a different name.
- Looking at the hyper-parameters in table 3, the number of layers and number of layers that do not share weights both equal 8 which means the module weights $W$ are not shared across layers. If this is the case, the sharing of signatures and codes across layers is an unnecessary constraint on the model (since modules in different layers perform different computations). Furthermore, this means the modules in different layers are completely different and the notion of a state at different layers does not make sense for the model.
- Considering the model’s similarity with PerceiverIO, the input is a MxC array of data that is transformed into a NxD latent array using the processor. PIO performs transformer-based attention on the N vectors before performing transforming them with an MLP. In NAC, SKMDPA is performed over U modules. It’s unclear how vectors of the latent array are channeled through modules. Can several vectors be processed by the same module ? And can several modules process the same vector ? It would be better to include the latent array dimension N in the description of the model and to clarify in detail the changes made to PIO to arrive at NAC.
- What is the architecture of the circuit generator ?
- How are CLIP text embeddings fed to the circuit generator ? How are all embeddings fed to PIO in the comparison experiment ? If both PIO and NAC achieve 64% accuracy on NLVR2 then NAC does not improve over PIO in this task.

---

> ### Author Response · Authors · 2022-08-01
> **Author Response to Reviewer g8fa [Part 1]**
>
> Thank you for reviewing our work and providing helpful feedback. We appreciate your view that NACs are a “novel method for modularizing neural networks”,  and a “flexible and efficient architecture” that “improves learning in low data regimes, OOD robustness and few-shot adaptation”. Your feedback has encouraged us to evaluate NACs on a wider range of benchmark tasks to strengthen the evidence for our claim that NACs are a general purpose modular architecture.
>
> In what follows, we address your concerns in the order they were presented.  To summarize, we:
>
> 1. Added experiments on 3 more modalities (point-cloud classification, symbolic processing and text classification from raw bytes), where we confirm that NACs are competitive general purpose models.
> 2. Clarified the comparison with Perceiver IO’s IID validation performance on ImageNet; the PIO baseline is a 48-layer deep model while our NAC is only 8 layers deep due to our resource constraints.
> 3. Improved the draft text to better communicate some of our model’s architectural details.
>
>
> ### Empirical results that demonstrate NACs are a General Purpose architecture
>
> We have broadened the scope of our experiments section to address the primary weaknesses as assessed by your review. Our new experiments span diverse data modalities (see below) while outperforming full-attention transformers and linear attention Linformers.
>
> The new modalities we investigate in addition to our original experiments on images and images + text are (a) ModelNet40 point clouds, (b) ListOps symbolic processing, and (c) IMDB text classification from a raw byte-stream (i.e., without text tokenization). Where applicable, we cite where the results are collected from the literature or specify that we performed the evaluations.
>
> **Point clouds.** A DeepMind team trained the Perceiver family of models on the point cloud classification benchmark ModelNet40 without additional data augmentation, geometric inductive biases or self-supervised pre-training [6]. In that setting, Perceiver IO obtained 77.4%, Hierarchical Perceivers achieved 75.9% without self-supervised pre-training, and when trained with self-supervised pre-training Hierarchical Perceivers obtained 80.6% (cf. Table 7 of [6]). We ran a comparable NAC without specialized tokenization or pre-training and achieved an 83% test accuracy. It appears that the sparse inductive prior was quite effective in this setting.
>
> | Method | Test Accuracy |
> | --- | --- |
> | Hierarchical Perceiver (No MAE) [6] | 75.9% |
> | Perceiver IO [6] | 77.4% |
> | Hierarchical Perceiver (With MAE) [6] | 80.6% |
> | **NAC (ours)** | **83.0%** |
>
> [6] Hierarchical Perceivers. [https://arxiv.org/abs/2202.10890](https://arxiv.org/abs/2202.10890)
>
> **ListOps Symbolic Processing.** The ListOps symbolic processing task is a challenging 10-way classification task on sequences of up to 2,000 characters. This commonly used benchmark [7] tests the ability of models to reason hierarchically while handling long contexts. The results are as follows:
>
> | Method | Test Accuracy |
> | --- | --- |
> | Full-Attention Transformers [7] | 37.13% |
> | Linformer [7] | 37.38% |
> | Perceiver IO (our impl.) | 39.70% |
> | **NAC (ours)** | **41.40%** |
>
> [7] Long Range Arena. [https://github.com/google-research/long-range-arena](https://github.com/google-research/long-range-arena)
>
> **Text Classification from Raw Bytes.** The IMDB task is a binary classification of sentiment on sequences of up to 4,000 bytes. In particular, we note that the models ingest raw ascii bytes, and **not** tokenized text.  All reported methods are trained without pre-training or data augmentation. The results are as follows:
>
> | Method | Test Accuracy |
> | --- | --- |
> | Full-Attention Transformers [7] | 65.35% |
> | Linformer [7] | 56.12% |
> | Perceiver IO (our impl.) | 66.50% |
> | **NAC (ours)** | **68.18%** |
>
> In addition to these experiments, we remark that the ImageNet few-shot results reported in Figure 4 also compares against a pre-trained Perceiver IO that has been trained by Jaegle et al. 2021 [8], the authors of Perceiver IO.
>
> Finally, we would like to note that we do not possess the computational resources or infrastructure to evaluate NACs on the same suite of benchmarks as Perceiver IO (which involves large-scale pretraining on hundreds of TPUs and potentially complex experimental infrastructure around Starcraft2).
>
> [8] Perceiver IO. [https://arxiv.org/abs/2107.14795](https://arxiv.org/abs/2107.14795)
>
>
> ---
> **Continued in Part 2**

---

> > ### Author Response · Authors · 2022-08-01
> > **Author Response to Reviewer g8fa [Part 2]**
> >
> > ### Large Scale Training and ImageNet Results
> >
> > As you point out, it is true that NAC lags behind Perceiver IO in terms of absolute IID validation accuracy on ImageNet, but there are two aspects that warrant consideration.
> >
> > First, we compare our NAC trained on ImageNet with a Perceiver IO baseline trained by the authors of the paper that propose it [3]. This ensures that the PIO baseline is well tuned, and minimizes the chances of errors that might result in the baseline not being adequately performant. However, this precludes a fair comparison with respect to IID performance since the PIO baseline is a 48-layer deep model while our NAC is only 8 layers deep due to our resource constraints.
> >
> > Second, recall that NACs implement modular inductive biases. This buys us considerably better few-shot adaptation, as conceptually argued in Bengio et al. 2019 [1] and empirically corroborated in Figure 4: NACs outperform Perceiver IO, **despite** the latter outperforming the former in terms of IID performance. In addition, we also see improved out-of-distribution performance, as shown in Table 1. However, there is no free lunch [2] and this inductive bias comes at a price: the in-distribution performance is negatively affected, since it becomes less favourable to arrive at solutions that overfit to the training distribution.
> >
> > [1] Bengio et al. 2019: A Meta-Transfer Objective for Learning to Disentangle Causal Mechanisms. [https://openreview.net/forum?id=ryxWIgBFPS](https://openreview.net/forum?id=ryxWIgBFPS)
> >
> > [2] Wolpert 1996: The Lack of A Priori Distinctions Between Learning Algorithms. [https://web.archive.org/web/20161220125415/http://www.zabaras.com/Courses/BayesianComputing/Papers/lack_of_a_priori_distinctions_wolpert.pdf](https://web.archive.org/web/20161220125415/http://www.zabaras.com/Courses/BayesianComputing/Papers/lack_of_a_priori_distinctions_wolpert.pdf)
> >
> > [3] Jaegle et al. 2021: Perceiver IO. [https://arxiv.org/abs/2107.14795](https://arxiv.org/abs/2107.14795)
> >
> > ### Identity of Modules Across Layers
> >
> > > […] the sharing of signatures and codes across layers is an unnecessary constraint on the model (since modules in different layers perform different computations). […]
> > >
> >
> > In a nutshell: while the sharing of codes across layers is optional (although not sharing codes negatively affects performance), the sharing of signature is not. Not sharing the signature fatally undermines the inductive bias of sparse communication between modules, as we elaborate below.
> >
> > Consider two modules A and B, that are constrained in their communication because their respective signatures are far enough away from each other across all the layers (which can be enforced by the graph prior regularizer). However, if we permit different signatures in each layer, it is entirely possible for their respective signatures at layer $l$ to be far away from each other (which would satisfy the regularizer), but then closer together in layer $l + 1$ (which can still satisfy the regularizer, if other modules are constrained in their place). In other words, modules A and B might be constrained in communicating at layer $l$, but it is possible that this constraint is lifted at layer $l + 1$. This implies any two modules can learn to communicate by “skipping a layer”, thereby bypassing the inductive bias of sparse communication between modules.
> >
> > This is less crucial for codes, but it is still conceptually important. A code defines the *identity* of a module over the layers, in that it couples the computation performed by that modules at all layers. Consequently, if a code is updated (say by gradient descent), so is the computation performed by that module at all layers. We did in fact experiment with this: we found that not sharing codes between layers lead to 5% lower IID accuracy and 8% lower OOD accuracy on Tiny-ImageNet and Tiny-ImageNet-R, relative to the baseline.
> >
> > Further, we also experimented with sharing all weights between propagator (layers), as well as sharing just the FFN weights but not the attention mechanism weights. For the model sizes that we were experimenting with, we found that the former lead to severe underfitting, whereas latter lead to moderate underfitting.
> >
> > Thank you for raising this important point, and we hope that we clarified the use of the notion of “state”.
> >
> > ---
> > **Continued in Part 3**

---

> > > ### Author Response · Authors · 2022-08-01
> > > **Author Response to Reviewer g8fa [Part 3]**
> > >
> > > ### Clarification of SKMDPA over Modules
> > >
> > > > Can several vectors be processed by the same module ? And can several modules process the same vector ?
> > > >
> > >
> > > The notion of a module in NACs generalizes that of a latent vector (a row in the latent array) defined in Perceiver IO [3]. Recall from Section 2 (Circuit Design) that each module in NACs can be described by three vectors: a signature vector, a code vector, and an initial state. The *initial state* exactly corresponds to a latent vector in Perceiver IO. Therefore, using your notation, $U$ (the number of modules) = $N$ (the number of latents). However, we note that there is no equivalent of signature and code vectors in Perceiver IO.
> > >
> > > In this context, SKMDPA in NACs is functionally a drop-in replacement for the transformer-based attention in Perceiver IO. Like transformer-based attention, it also consumes $N = U$ input vectors and outputs $N = U$ output vectors. Unlike transformer-based attention where all input vectors can influence all output vectors, only some input vectors can influence some other output vectors in SKMDPA. Which vector influences which other vector is specified by the corresponding signatures (recall that there are exactly $N = U$ of these), which are also learned.
> > >
> > > Thank you for asking this, it helped us improve the clarity of the manuscript.
> > >
> > > ### Architecture of the Conditional Circuit Generator
> > >
> > > Thank you for pointing out this omission, we will include a description in the main text. In our experiments with conditional NACs, the circuit generator was implemented by a simple cross attention layer followed by a feed forward network (FFN), and a self-attention layer followed by two FFNs run in parallel (more on that below). In the initial cross-attention layer, the keys and values were derived by linearly transforming pre-pooling CLIP text embeddings (which yields one embedding vector per text token). We used CLIP-base (the smallest available pre-trained CLIP) to minimize the compute requirement. The queries in the first cross-attention layer were learned 384-dimensional vectors, of which we had exactly $U$ (i.e. one per module). The output of the self attention layer was therefore another set of $U$ vectors, each of which was passed through two different FFNs in parallel. The first FFN produced the signatures for each module ($U$ 64-dimensional vectors), whereas the second FFN produced the corresponding codes ($U$ 384-dimensional vectors).
> > >
> > > In conclusion, we would like to reiterate that we found this review helpful and that we believe this rebuttal process has resulted in significant improvements to our submission, particularly in the empirical results. Thank you again for this contribution, and please feel invited to engage with us in case you have further questions.

---

> > > > ### Comment · Reviewer_g8fa · 2022-08-07
> > > > **Score increase**
> > > >
> > > > I thank the authors for the thorough replies. They address all the concerns that I raised and they provide important clarifications of the model architecture. As long as these changes are incorporated in the final version of the paper, I will recommend accepting the paper and increase the score.

---

> > > > > ### Author Response · Authors · 2022-08-08
> > > > > **Thank you.**
> > > > >
> > > > > Thank you for your interacting with us, and we appreciate that you have adjusted your score given the additional results and clarifications.
> > > > >
> > > > > We will incorporate the resulting changes in the final revision, which will grant us an extra page to work with (updates during the rebuttal period are unfortunately still subject to the 9-page limit).

---

> ### Author Response · Authors · 2022-08-06
> **Thank you again for your review.**
>
> We thank you again for your positive and constructive review. We were glad to hear that you found our work to be novel, and our architecture to have a *flexible and efficient* design.  Your questions on the identity of modules, SKMDPA and the design of the circuit generator have helped us improve our presentation as we prepare the next full revision.
>
> Your primary concern was around our claim that NACs are general purpose, given that the submission version only presented experiments in the image and image + text domain. Based on this feedback, we have conducted additional experiments on a selection of three diverse domains including 3D point-clouds, symbolic processing, and text classification from raw bytes (i.e. without tokenization). These new results show that NACs are indeed general purpose and compare favourably with existing baselines in the literature (including vanilla transformers, linear attention transformers, and existing general purpose architectures).
>
> We believe that these new results and improvements resulting from your questions have significantly strengthened our work. If you hold the same opinion, we would appreciate it if you would consider updating your score to reflect this. If not, please feel invited to interact with us. Thank you.

---

### Official Review · Reviewer_myJN · 2022-07-21

**Rating:** 6
**Confidence:** 4
**Soundness:** 3 good
**Presentation:** 3 good
**Contribution:** 3 good

**Summary:**

This work proposes Neural Attentive Circuits (NAC) which learns module parameterization and the sparse connectivity between modules trained jointly End-to-end. The architecture is based on the Perciever-IO arhictecture which serves as a baseline for their ablation experiments. They introduce a SKMDPA which is a sparse attention mechanism utilizing with the Concrete distribution (Gumbel Softmax Distribution) and ModFC and ModFFN which conditions their compution on an input code. These modules control the attention connectivity and their computation based on the output of the Circuit Generator which can either be input-conditional or unconditional. They demonstrate the input-uncondition version of their work on a few-shot image classification tasks on the Cifar and CUB datasets and achieve better results than their baseline Perciever network.  They show with ablation on tiny imagenet that simply adding their modules and learnable sparse connectivity does not significantly improve in distirbution results, but greatly improve out of distribution results. They further demonstrate adding different graph-connectivity priors to the learned connectivity also significantly improves performance. They experimented with the sample-conditioned circuit design generation trained on the Natural Language for Visual Reasoning for Real using the CLIP encoder to preprocess both text and images. The circuit generator is conditioned only on the input text and only the image is given to the network executor. They visualize a TSNE of the connectivity graphs output by the circuit generator and show that their is a strong grouping in graph connectivity based on type of question given.

**Questions:**

Can you compare your model in the same setting as the Perciever-IO paper? Do you have metrics for comparison on language tasks? It would make it more convincing to compare this model as a general perpose architecture if there comparisons in more modalities.

It is likely useful to compare this work to other works for sparse attention and dynamic routing to increase inference efficiency as well as NAS works which also use the concrete distribution and other hard selection mechanisms for architecture search. [1], [2], [3].

The uncondition circuit generator work is significantly similar to many works in NAS and sparsification and this work would benefit greatly from comparison.

The Conditional Circuit Generator is quite novel especially if applicable to multimodal data, but the performance and properties of the network could be explored in much more detail.

Could you elaborate about the ModFNN properties?
"unlike in transformers, each copy is conditioned by a unique learnable code vector and therefore performs a different computation. Consequently, though the computation performed in each module is different, the total number of parameters does not noticeably increase with the number of modules"
I'm not sure I understand. Are they unique and increases parameters or are they shared? Some of it is elaborated on in section 2.2, but it still not entirely clear if they are unique to each layer or module.

Can accuracy and performance be discussed in section 4.2 with baselines? It would allow stronger conclusions be ot made about the results.

For "We observe that the Neural Attentive Circuit (NAC) is much more robust than Perceiver IO to sparsification at inference time" do you have results with PercieverIO sparsification baselines?

For Figure 5b, do you have a baseline for the compute cost and inference speed of Perciever IO?

May be useful to compare to [5]. The results with the meta-network inference such as in section 6.3 is somewhat similar to the work with an Unconditional Circuit Generator.

Unrelated to the paper quality, but could this architecture be effective for continual learning? Some methods in continual learning also try to dynamically change the network to partition off tasks to prevent catastrophic forgetting.

[1] Cai, Shaofeng, Yao Shu, and Wei Wang. "Dynamic routing networks." Proceedings of the IEEE/CVF Winter Conference on Applications of Computer Vision. 2021.
[2] Xie, Sirui, et al. "SNAS: stochastic neural architecture search." International Conference on Learning Representations. 2018.
[3] Correia, Gonçalo M., Vlad Niculae, and André FT Martins. "Adaptively Sparse Transformers." Proceedings of the 2019 Conference on Empirical Methods in Natural Language Processing and the 9th International Joint Conference on Natural Language Processing (EMNLP-IJCNLP). 2019.
[4] Yoon, Jaehong, et al. "Lifelong Learning with Dynamically Expandable Networks." International Conference on Learning Representations. 2018.
[5] Shaw, Albert, et al. "Meta architecture search." Advances in Neural Information Processing Systems 32 (2019).

**Limitations:**

I'm not sure the authors should claim this is a general-purpose modular neural network architecture without experiments in multiple modalities.

**Strengths And Weaknesses:**

Strengths:
- Proposes a novel architecture for jointly learning the sparse connectivity between modules and the model parameters.
- Ran ablation which demonstrated their improvements with learned non sample-conditioned sparse connectivity can improve out of distribution accuracy on tiny-imagenet and further graph priors improve in and out of distribution accuracy.
- Demonstrated improvements over Perciever-IO on few-shot adaptation.
- Demonstrated with a visualization that the architecture can learn different paths for different types of problems on the Natural Language for Visual Reasoning for Real datset.
- Demonstrated that their method can be used to sparsify the architecture and achieve the same performance as Perciever-IO with 80% of circuits dropped. They further showed that this leads to a real inference speedup.

Weaknesses:
- Does not demonstrate the architecture as a General purpose architecture after introducing it as such. The results with the text modality use clip embeddings.
- Weak empirical results. No strong baselines for the non sample dependent results. Very similar to architecture search and sparsification, but no reference. No comparisons on the same tasks as the original Perciever-IO paper.
- The sparsification results seem strong, but still there aren't comparisons to other sparsification methods or Neural Architecture search methods.

---

> ### Author Response · Authors · 2022-08-01
> **Author Response to Reviewer myJN [Part 1]**
>
> Thank you for the effort you have invested in reviewing our work. We are glad that you found our proposed architecture novel and that its modular inductive bias can improve in- and out-of-distribution accuracy on several tasks. Your perceptive critical feedback has helped us to extend and refine the paper. In what follows, we provide a detailed response to your questions and comments.
>
> **In summary, we:**
>
> 1. Added experiments on 3 more modalities (point-cloud classification, symbolic processing, and text classification from raw bytes) that confirm NACs are competitive general purpose models.
> 2. Strengthened our empirical results by comparing against the baselines that you mention [1, 2, 5] where they overlap with our tasks. We found that NACs compare favourably.
> 3. Elaborated on some of the details around ModFFNs and sparsification.
>
> ### Empirical Results that Demonstrate NACs are a General Purpose Architecture
>
> We have broadened the scope of our experiments section to address the primary weaknesses as assessed by your review. Our new experiments span diverse data modalities (see below) while outperforming full-attention transformers and linear attention Linformers.
>
> The new modalities we investigate in addition to our original experiments on images and images + text are (a) ModelNet40 point clouds, (b) ListOps symbolic processing, and (c) IMDB text classification from a raw byte-stream (i.e., without text tokenization). Where applicable, we cite where the results are collected from the literature or specify that we performed the evaluations.
>
> **Point clouds.** A DeepMind team trained the Perceiver family of models on the point cloud classification benchmark ModelNet40 without additional data augmentation, geometric inductive biases or self-supervised pre-training [6]. In that setting, Perceiver IO obtained 77.4%, Hierarchical Perceivers achieved 75.9% without self-supervised pre-training, and when trained with self-supervised pre-training Hierarchical Perceivers obtained 80.6% (cf. Table 7 of [6]). We ran a comparable NAC without specialized tokenization or pre-training and achieved an 83% test accuracy. It appears that the sparse inductive prior was quite effective in this setting.
>
> | Method | Test Accuracy |
> | --- | --- |
> | Hierarchical Perceiver (No MAE) [6] | 75.9% |
> | Perceiver IO [6] | 77.4% |
> | Hierarchical Perceiver (With MAE) [6] | 80.6% |
> | **NAC (ours)** | **83.0%** |
>
> [6] Hierarchical Perceivers. [https://arxiv.org/abs/2202.10890](https://arxiv.org/abs/2202.10890)
>
> **ListOps Symbolic Processing.** The ListOps symbolic processing task is a challenging 10-way classification task on sequences of up to 2,000 characters. This commonly used benchmark [7] tests the ability of models to reason hierarchically while handling long contexts. The results are as follows:
>
> | Method | Test Accuracy |
> | --- | --- |
> | Full-Attention Transformers [7] | 37.13% |
> | Linformer [7] | 37.38% |
> | Perceiver IO (our impl.) | 39.70% |
> | **NAC (ours)** | **41.40%** |
>
> [7] Long Range Arena. [https://github.com/google-research/long-range-arena](https://github.com/google-research/long-range-arena)
>
> **Text Classification from Raw Bytes.** The IMDB task is a binary classification of sentiment on sequences of up to 4,000 bytes. In particular, we note that the models ingest raw ascii bytes, and **not** tokenized text.  All reported methods are trained without pre-training or data augmentation. The results are as follows:
>
> | Method | Test Accuracy |
> | --- | --- |
> | Full-Attention Transformers [7] | 65.35% |
> | Linformer [7] | 56.12% |
> | Perceiver IO (our impl.) | 66.50% |
> | **NAC (ours)** | **68.18%** |
>
> In addition to these experiments, we remark that the ImageNet few-shot results reported in Figure 4 also compares against a pre-trained Perceiver IO that has been trained by Jaegle et al. 2021 [8], the authors of Perceiver IO.
>
> Finally, we would like to note that we do not possess the computational resources or infrastructure to evaluate NACs on the same suite of benchmarks as Perceiver IO (which involves large-scale pretraining on hundreds of TPUs and potentially complex experimental infrastructure around Starcraft2).
>
> [8] Perceiver IO. [https://arxiv.org/abs/2107.14795](https://arxiv.org/abs/2107.14795)
>
> ----
> **Continued in Part 2**

---

> > ### Author Response · Authors · 2022-08-01
> > **Author Response to Reviewer myJN [Part 2]**
> >
> > ### Similarity to Neural Architecture Search and Dynamic Routing
> >
> > Thank you for the references to these works. Neural Architecture Search (NAS) is indeed related, and we have included a discussion in the related work. In summary: many neural architecture search methods leverage learned routing of information through the network similar to NACs, but the raison d’être of neural architecture search is quite different as we explain below.
> >
> > A key objective of NAS is to search over a variety of *heterogenous* (different) architectural primitives, e.g., different activation functions, layer types, etc. In NACs, however, our objective is to compose a set of *homogenous* (identical) primitives (programmable modules) in a flexible yet learned way. Further, we are not aware of a work that applies NAS to obtain a general purpose architecture like ours. That said, we note that the use of heterogenous primitives is compatible with SKMDPA, and learning a circuit design over such primitives is an exciting avenue of future work, but that is beyond the scope of the current submission.
> >
> > Finally, we include comparisons with some of the works that you suggest which have tasks that overlap with ours. However, we also note that these baseline models are domain-specific, while NACs are general purpose.
> >
> > Re: [1] — NACs compare favourably against DRNet(L) on ImageNet: 77% top-1 accuracy (NACs) vs. 70.2% top-1 accuracy (DRNet(L)).
> >
> > Re: [2] — NACs compare favourably against SNAS on ImageNet: 77% top-1 accuracy (NACs) vs. 72.7% top-1 accuracy (SNAS).
> >
> > Re [5] — NACs compare favourably against BASE on ImageNet: 77% top-1 accuracy (NACs) vs. 74.3% top-1 accuracy (BASE ImageNet Tuned).
> >
> > Re [3] — There are no overlapping tasks that we can compare with.
> >
> > ### Properties of ModFFNs
> >
> > Thank you for asking this — we will expand the discussion in Section 2.2 in the next revision.
> >
> > A large chunk of ModFFN parameters is shared between modules, but a small number of parameters (the code vectors, in the unconditional case) is not. Let’s consider a concrete example — a 2 layer deep ModFFN that takes in $d_{in} = 384$ dimensional inputs and returns $d_{out} = 384$ dimensional outputs, with a hidden layer size of $d_{h} = 1536$. Let us assume that the number of modules is $U$, each associated with a code vector of dimension $d_c = 384$. Let us further assume that the total number of such ModFFNs in the network is $L = 8$. The total number of parameters in all ModFFNs is given by:
> >
> > $$
> > L \cdot (2 \cdot d_{in} \cdot d_{h} + 2 \cdot d_{c} \cdot d_{h}) + U\cdot d_{c} = 1.89 \times 10^7 + 384 \cdot U
> > $$
> >
> > The multiplier 2 is due to the fact that each FFN has two layers. We can observe that the number of parameters increases only very modestly with the number of modules $U$. It would take more than 10000 modules before the contribution due to code vectors starts becoming noticeable.
> >
> > ### Conditional Circuit Generator
> >
> > One exciting aspect of the NAC architecture is that it can perform conditional computation. We agree that this direction could be explored in much greater depth, but our goal in Section 4.2 was to show a proof-of-concept. Indeed, we showed that input sentences found in NLVR2 requiring different skills (e.g., measuring hard cardinality, spatial associations, etc.) result in different types of circuit designs. However, NLVR2 is a small dataset comprising roughly 100k samples, and most state-of-the-art methods that quantitatively benchmark on it (e.g., [9]) rely on large-scale pre-training on vision language tasks in order to not overfit. This limits the extent to which we can draw quantitative insights from this dataset without large-scale pretraining, for which we do not have access to the required compute nor the belief that it is a necessary pre-condition to demonstrate in general the efficacy of our model.
> >
> > [9] CoCa: Contrastive Captioners are Image-Text Foundation Models. [https://arxiv.org/abs/2205.01917](https://arxiv.org/abs/2205.01917)
> >
> > ---
> > **Continued in Part 3**

---

> > > ### Author Response · Authors · 2022-08-01
> > > **Author Response to Reviewer myJN [Part 3]**
> > >
> > > ### Perceiver IO Baseline for Sparsification
> > >
> > > In response to your questions about compute cost and inference speed of Perceiver IO, we have created a new figure to clarify the effect of sparsification on NACs (with different graph priors) and Perceiver IO. The figure shows on the x-axis shows GFLOPs / sample, while on the Y-axis is Tiny-ImageNet validation accuracy. We vary the GFLOPs by dropping modules in NACs or latents in Perceiver IO.
> > >
> > > [link to anonymized figure](https://sevn.s3.us-east-1.amazonaws.com/floppy_bird.pdf)
> > >
> > > We make two observations:
> > >
> > > 1. For NACs, even when the FLOP budget is reduced by a factor of 5 (to roughly 1.6 GFLOPs), the amount of performance lost is remarkably small, corresponding to e.g., roughly 1% in accuracy difference w.r.t the full model with ring-of-cliques prior. In contrast, Perceiver IO loses 5.3% in validation accuracy w.r.t the full model when given roughly the same amount of FLOPs (1.7 GFLOPs).
> > > 2. Under 1 GFLOP, Perceiver IO degrades slower than NACs. This is because the cross attention in Perceiver IO is computationally lighter than the read-in attention in NACs. This difference only starts becoming visible in the very low compute regime (i.e., under 1 GFLOP), where the read-in attention not amortized by the other attention mechanisms and FFNs. We further note that the read-in attention in NACs can be replaced by the computationally cheaper cross attention like in Perceiver IO, at the cost of a modest amount of performance (corresponding to roughly 1% validation accuracy on ImageNet).
> > >
> > >
> > > ### Opportunities for Continual Learning
> > >
> > > We believe that applying NACs to continual learning is perhaps one of the most exciting directions for future work, and an important direction for the field of machine learning.  NACs are a sparse modularization of a general-purpose model, and as we show in section 4.2 we are able to learn input conditional circuit designs. As you point out, this could be useful in the prevention of catastrophic forgetting.
> > >
> > > In summary, we thank you again for your review. We would be eager to interact with you in case you have additional questions.

---

> > > > ### Comment · Reviewer_myJN · 2022-08-07
> > > > **Response to Rebuttal**
> > > >
> > > > I would like to first thank the author's greatly for their consideration and significant work for additional experiments and analysis. I had some short questions and comments I wanted to ask before the end of the discussion period.
> > > >
> > > > >Finally, we include comparisons with some of the works that you suggest which have tasks that overlap with ours. However, we also note that these baseline models are domain-specific, while NACs are general purpose.
> > > >
> > > > Did you happen to have FLOP and parameter count comparisons?
> > > >
> > > > >We make two observations:
> > > >
> > > > >For NACs, even when the FLOP budget is reduced by a factor of 5 (to roughly 1.6 GFLOPs), the amount of performance lost is remarkably small, corresponding to e.g., roughly 1% in accuracy difference w.r.t the full model with ring-of-cliques prior. In contrast, Perceiver IO loses 5.3% in validation accuracy w.r.t the full model when given roughly the same amount of FLOPs (1.7 GFLOPs).
> > > > >Under 1 GFLOP, Perceiver IO degrades slower than NACs. This is because the cross attention in Perceiver IO is computationally lighter than the read-in attention in NACs. This difference only starts becoming visible in the very low compute regime (i.e., under 1 GFLOP), where the read-in attention not amortized by the other attention mechanisms and FFNs. We further note that the read-in attention in NACs can be replaced by the computationally cheaper cross attention like in Perceiver IO, at the cost of a modest amount of performance (corresponding to roughly 1% validation accuracy on ImageNet).
> > > >
> > > > Do you happen to have the sparsification results for your model with the cheaper cross attention? What is the main reason for the read-in attention if it can be replaced in your model?

---

> > > > > ### Author Response · Authors · 2022-08-08
> > > > > **Additional Clarifications**
> > > > >
> > > > > Thank you for your follow-up questions.
> > > > >
> > > > > > Did you happen to have FLOP and parameter count comparisons?
> > > > > >
> > > > >
> > > > > Yes, we have collected FLOP and parameter counts from the related literature:
> > > > >
> > > > > | Model | GFLOPs (G) | Parameter Count (M) |
> > > > > | --- | --- | --- |
> > > > > | NACs | 8.16 | 35.8  |
> > > > > | DRNet [1] | 0.60 | 4.4 |
> > > > > | SNAS [2] | 0.52  | 4.3 |
> > > > > | BASE [5] | Not reported | 3.3 |
> > > > >
> > > > > However, as we have discussed, a direct comparison would be misleading for two main reasons:
> > > > >
> > > > > 1. The neural architecture search in [1], [2] and [5] is over architectures that are domain specific. More precisely, these models only work for images whereas NACs are general-purpose. Because NACs have fewer (i.e. no) domain-specific inductive biases than DRNet [1], SNAS [2], and BASE [5], it follows that NACs can leverage more compute and parameters to yield better performance.
> > > > > 2. The computational pipeline in NAS is specialised and often involves GPU days of search time (e.g. 8 days in BASE [5], 1.5 days in SNAS [2] and 1 day in DrNet [1]). Since training a NAC does not involve architecture search, there is no compute overhead due to search time; instead, the model is trained like any other classification model, e.g. a ResNet or Perceiver IO.
> > > > >
> > > > > Thank you for asking this question, we will include a discussion about these details in our next revision.
> > > > >
> > > > > > Do you happen to have the sparsification results for your model with the cheaper cross attention? What is the main reason for the read-in attention if it can be replaced in your model?
> > > > > >
> > > > >
> > > > > Perceiver-style models have used two different kinds of read-in cross-attention. The original Perceiver used multiple iterations of cross attention between the latent array and the input array at different layers, whereas the Perceiver IO model used only a single cross attention between the latent array and input array to reduce the computational cost. During our experimentation we tried Perceiver IO-style cross attention as well as the novel approach (read-in attention) included in our main paper. We found that the Perceiver IO-style cross attention performed 1% worse on ImageNet top-1 accuracy w.r.t the read-in attention, while only moderately improving the run-time (by roughly 10%). We decided to keep the performance gain resulting from using our read-in attention. We do not have the sparsification results for the NAC with Perceiver-IO style cross-attention.
> > > > >
> > > > > We believe that our additional experiments and that this discussion has strengthened our work with respect to the primary weaknesses outlined in your review. If you share this perspective, we hope that you would reconsider your rating of this paper.

---

> ### Author Response · Authors · 2022-08-06
> **Thank you again for your review.**
>
> We thank you again for your detailed and constructive review. We were pleased to learn that you found our work novel and that you appreciated the study of NAC’s interesting properties. Your numerous questions have helped us improve the presentation as we prepare a new revision.
>
> Your primary concern was that our claim that NACs are a general purpose architecture was not well-supported. Based on your feedback, we have run a suite of additional experiments on three diverse data modalities. We found NACs to excel in each of these, also when compared against existing baselines in the literature (including vanilla transformers, linear attention transformers, and existing general purpose architectures). In the domain of 3D point-clouds, we directly compare NACs with known results for Perceiver IO, while noting that we do not possess the compute resources or experimental infrastructure to evaluate on the same set of experiments as Perceiver IO (which involves large scale training on hundreds of TPUs). Additionally, as you requested, we included additional results that compare the sparsification properties of NACs and Perceiver IO, where we found NACs to offer a competitive compute-performance trade-off.
>
> We believe that these additional results significantly strengthen our work. If you hold the same opinion, we would appreciate it if you would update your rating of our submission to reflect this. If not, we invite you to interact with us. Thank you.

---

### Author Response · Authors · 2022-08-01
**General Comment to All Reviewers**

We are grateful to all reviewers for the time and effort they have invested in reviewing our work and we provide a detailed response to each review as a direct comment. We are pleased to see that all reviewers agreed on the value of developing general purpose models with modular inductive biases. In addition, that they all found our proposed architecture “novel” (Reviewers myJN, g8fa, p9jH), having “a flexible and efficient architecture design” (Reviewer g8fa), and that “the conditional circuit generator is quite novel” (Reviewer myJN). We are also glad that reviewer p9jH finds the “writing is clear and easy to follow” and that “the figures are well made and very helpful”.

The reviewers generally appreciate the experiments that we present. However, their primary concern was that our claim that Neural Attentive Circuits (NACs)  are general purpose models was not adequately supported, given that we only provide experiments in the image and image + text domain. To alleviate this concern, we have conducted additional experiments on a selection of three diverse domains: (a) 3D point-clouds (ModelNet40), (b) symbolic processing (ListOps) and (c) text classification from raw ascii bytes (i.e. without text tokenization). **We found NACs to excel in these domains**, outperforming general purpose and transformer baselines reported in the literature and clearly demonstrating the general purpose nature of NACs.

Finally, as we prepare the next revision, we are grateful the reviewers’ numerous questions have helped us identify where our presentation can be further improved.

---

### Meta-Review · Area_Chair_CYv6 · 2022-08-29

**Recommendation:** Accept
**Confidence:** Certain

**Metareview:**

The reviewers unanimously agreed that the work presented here was novel and improved the state of the art in models that perform conditional computation on a wide variety of different kinds of data. The reviewers were originally concerned that the submission focused evaluation only on visual data, but the authors provided additional data during the response period that made clear the generality of the proposed technique. As such, I recommend acceptance. I look forward to visiting the authors' poster in New Orleans! Authors: this enthusiastic acceptance is predicated on the expectation that you update the paper with these new results.

**Award:**

No

---

### Decision · Program_Chairs · 2022-09-14

Accept